# A computational account of why more valuable goals seem to require more effortful actions

**Emmanuelle Bioud\*, Corentin Tasu, Mathias Pessiglione\***

Motivation, Brain and Behavior lab, Paris Brain Institute (ICM); Sorbonne Université; Inserm U1127; CNRS U7225, Paris, France

**Abstract** To decide whether a course of action is worth pursuing, individuals typically weigh its expected costs and benefits. Optimal decision-making relies upon accurate effort cost anticipation, which is generally assumed to be performed independently from goal valuation. In two experiments (n = 46), we challenged this independence principle of standard decision theory. We presented participants with a series of treadmill routes randomly associated to monetary rewards and collected both 'accept' versus 'decline' decisions and subjective estimates of energetic cost. Behavioural results show that higher monetary prospects led participants to provide higher cost estimates, although reward was independent from effort in our design. Among candidate cognitive explanations, they support a model in which prospective cost assessment is biased by the output of an automatic computation adjusting effort expenditure to goal value. This decision bias might lead people to abandon the pursuit of valuable goals that are in fact not so costly to achieve.

## Editor's evaluation

This article tests a basic assumption made by many decision-making theories, specifically that costs and benefits are independent when making rational choices. As such, the findings from this study should be of interest to a wide range of researchers interested in decision-making, effort, and motivation.

**\*For correspondence:**
emmanuelle.bioud@gmail.com (EB);
mathias.pessiglione@gmail.com (MP)

**Competing interest:** The authors declare that no competing interests exist.

## Introduction

Should you walk to the nearest bakery or cycle to your favourite one? Should you climb a little higher to get a better view or just stay below and enjoy what you can see already? This kind of choice, about whether or not to trade an effort for a reward, has been investigated and modelled in neuro-economic studies. The general idea of decision-making theory is that the subjective value of the goal, defined as an anticipated outcome, is discounted by some function of the expected cost, related to the energetic expense (*Chong et al., 2017*; *Hull, 1943*; *Kable and Glimcher, 2009*; *Klein-Flügge et al., 2015*; *Mitchell, 2003*; *Rangel et al., 2008*; *Schmidt et al., 2012*). Crucially, these theories make the implicit assumption that individual judgements about goal value and effort cost are made independently from each other, before being integrated into a discounted value on which decision is based. More generally, most choice models assume that the different costs integrated in the discounted value computation (not only effort, but also delay, risk, etc.) are estimated separately, a principle known as the independence axiom.

Yet, it has been several decades since the first reports of an interplay between outcome value and effort cost judgements have appeared in the psychology literature (*Lewis, 1965*). One line of research has described numerous instances in which exerted effort inflates the perceived value of

the associated outcome (*Arkes et al., 1994*; *Aronson and Mills, 1959*; *Inzlicht et al., 2018*). Some authors have considered this phenomenon as an attempt to justify effort expenditure, following on a need to reduce cognitive dissonance (*Inzlicht et al., 2018*). For example, yielding to the so-called 'IKEA effect' (*Norton et al., 2012*), individuals tend to ascribe greater value to an object that they assembled themselves compared to one that was assembled by an expert. A distinct line of research has documented cases in which the value of the outcome modulates one's retrospective judgement about the amount of effort invested to reach it. For example, people tend to incorporate the monetary reward into their retrospective judgement of effort intensity using the positive correlation between reward and effort in a Bayes-optimal fashion (*Pooresmaeili et al., 2015*; *Rollwage et al., 2018*).

Thus, the foundational assumption of mutual independence between judgements of effort cost and outcome value appears at odds with a large body of empirical observations. However, all cases of mutual influence between effort and value reported in the literature were manifested retrospectively, once effort was already expended, and the outcome actually obtained. The question of whether prospective effort cost estimates might be modulated by the expected value of the outcome (i.e., goal value) has so far received little attention. Yet this seems an important question, given that decisions are obviously based on prospective, not retrospective, effort cost estimates.

The most widespread instance of systematic cost distortion is perhaps the 'planning fallacy', that is, the tendency observed in healthy agents to underestimate the time necessary to complete a task or a project (*Brunnermeier et al., 2008*; *Buehler et al., 1994*; *Kahneman and Tversky, 1979*), even when they have extensive experience about it. While this fallacy has mainly been reported in the context of delay estimation, it also occurs for other types of cost, such as money (*Peetz and Buehler, 2013*; *Peetz and Buehler, 2009*) and effort (*Morgenshtern et al., 2007*). The planning fallacy has been proposed to result from an excessive focus on the future goal, to the detriment of information accumulation about both the intermediate steps to be taken, and declarative or experiential knowledge about the task at hand (*Buehler and Griffin, 2003*; *Kruger and Evans, 2004*). This attentional bias could have been selected to favour engagement in strenuous actions by partially ignoring their costs. If more attractive goals capture more attention, then effort costs should be even more underestimated when the goal is assigned a higher value.

Therefore, our research aimed to (1) assess the potential impact of goal value on prospective effort cost judgements, and if present, (2) elucidate the cognitive mechanisms underlying this impact. To this end, we set up a novel experimental paradigm in which healthy participants prospectively appraised the energetic cost of various action sequences leading to a more or less valuable outcome. The action sequence was a run on a treadmill, composed of multiple segments with varying speed and slope, which was rewarded with a financial payoff of varying magnitude. Specifically, participants first viewed animations depicting a cartoon character running along a more or less effortful multi-segment route, and subsequently estimated the energetic cost of the route. The mapping between the visual display of running routes and their energetic cost was established prior to the main task by having participants first run routes of various speed and slope on the treadmill (practice session) and then learn to guess their energetic cost based on objective feedback (calibration session). During the main task, each route was explicitly paired with a monetary incentive, whose magnitude was varied orthogonally to energetic cost fluctuations. In order to trigger decision-related cost/benefit computations, participants were offered a choice between a high-effort high-value new route and fixed low-effort low-value reference route. We ensured that effort costs and monetary incentives had personal significance to the participants by having a randomly selected fraction of their choices implemented in reality, meaning that they were requested to perform the chosen action sequence on the treadmill and were given the associated payoff. In line with our planning fallacy hypothesis, we predicted that participants would anticipate a lower energetic cost for routes paired with a higher-value monetary outcome.

## Results
### Experiment 1

With this experiment, we intended to measure the impact of goal value onto the anticipated effort cost of an action sequence. We used composite running routes, each consisting of four segments of varying slope and speed, as the action sequences about which participants would make a priori energetic cost judgements. We manipulated the value attached to the completion of each route by

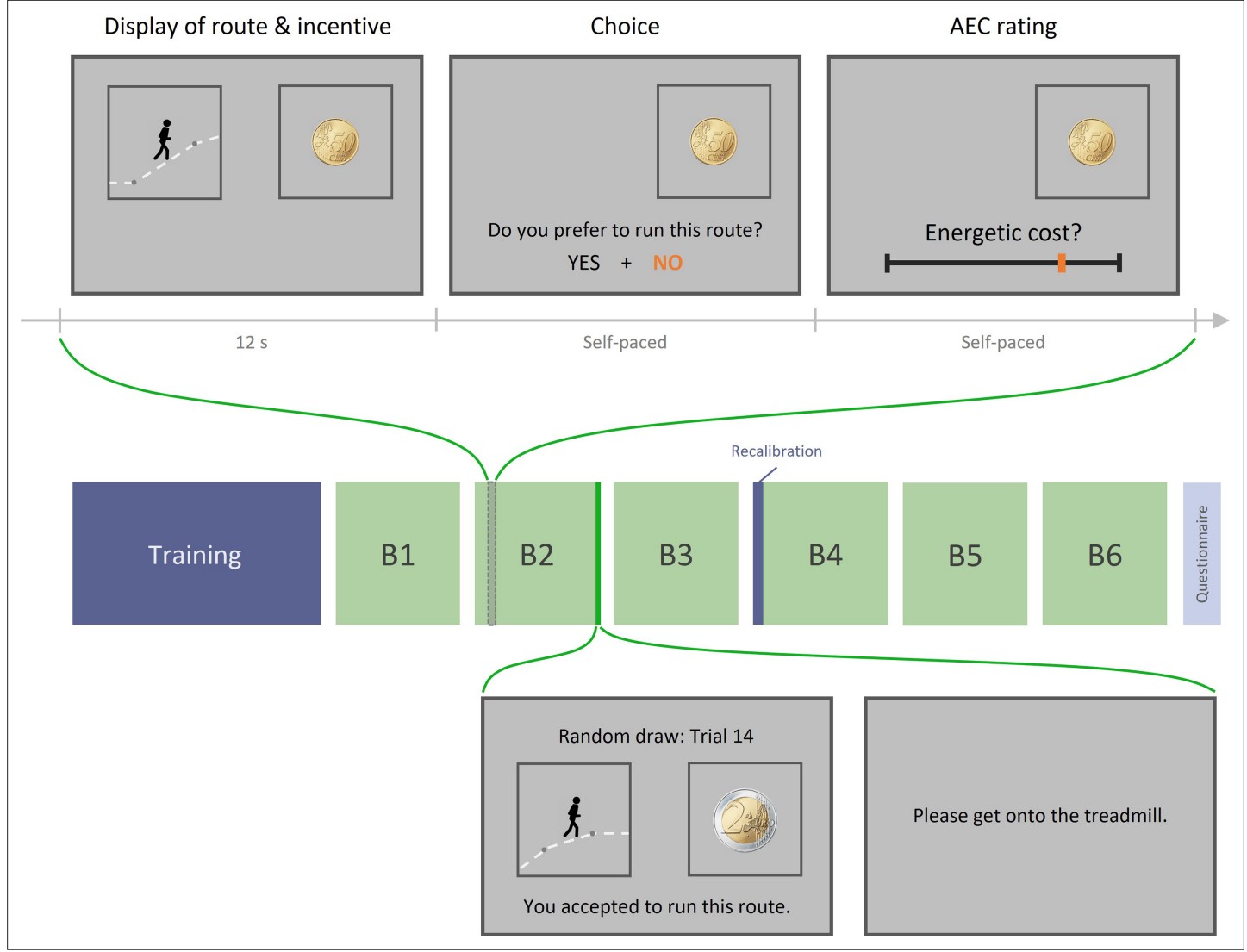

**Figure 1.** Experiment 1 design. Top row: example trial. Each trial includes three phases: (1) display of the running route animation and the associated incentive, (2) choice between this high-effort high-gain (HEHG) option or an implicit (not displayed) low-effort 0-gain (LE0G) option, and (3) rating of anticipated energetic cost (AEC). Middle row: structure of a session. Each session includes a training phase, three blocks of 18 trials, a recalibration of AEC rating, and three additional blocks. The training phase comprised a practice session (running various routes on the treadmill) and a calibration session (learning to provide accurate energetic cost rating based on objective feedback). The recalibration session only included a series of ratings with objective feedback about the real energetic cost. Bottom row: Random draw performed at the end of each block among its 18 just-completed trials. Choice made by the participant during the randomly selected trial is reminded and then executed: if the participant accepted the high-effort offer, she must get onto the treadmill and complete the effortful running route, after which the associated reward is added to her total earnings. If she declined the high-effort offer, she must complete a default low-effort running route, and her total earnings remain unchanged.

offering monetary rewards of varying magnitude (from 0.01€ to 5€). Specifically, in each trial, participants were first presented with the picture of a monetary amount (the incentive, or potential reward) on a computer screen, followed by a short animation depicting a cartoon character running along a computer-generated route (see *Figure 1*). Next, participants were required to choose between accepting this high-effort high-gain (or HEHG) new offer (namely, running this route on a treadmill in order to earn the displayed incentive) or discarding it in favour of a fixed low-effort 0-gain (or LE0G) reference option (namely, running a default route at participant's self-set comfort speed and 0% slope, for no reward). Lastly, they were prompted to estimate the energetic cost of this route, which was our main behavioural measure of interest, by placing a cursor along an analogue scale. Although this was not made explicit to the participants, the lower end of the scale corresponded to the energetic cost of a 2-min run at the easiest speed and slope levels used in route composition (110% of comfort speed,

0% slope), while the higher end of the scale corresponded to a 2-min run at the highest slope and speed levels (140% of comfort speed, 8% slope). For obvious ethical reasons, effort levels were thus kept in a limited range, such that participants remained confident that they would cope and not fall off the treadmill (which never happened in practice).

Of note, trials were grouped in blocks of 18, and once a block had been completed, one trial was randomly drawn from this block and executed: participants were asked to complete the selected running route (or the comfort route if they had declined the high-effort offer) on a treadmill, and the attached monetary reward was then added to their final payment. This procedure was intended to reinforce the personal relevance of the displayed reward magnitudes and of the energetic cost estimates made by our participants.

Critically, the reward magnitude and the real energetic cost (REC) of the route were orthogonal by design, so the incentive cue was in no way informative about the rating that participants should provide. Before the experiment, participants were trained during a calibration session to estimate the energetic cost of various running routes, whose objective value was computed using standard equations in the physical exercise literature (*Hall et al., 2004*). Participants were trained again with objective feedbacks during a recalibration session conducted between the two halves of the experiments. Also, it was in their interest to make sound judgements during the experiment as a monetary bonus could be earned on the basis of rating accuracy (see 'Materials and methods' for details). Thus, any impact of incentive level we might observe on the anticipated energetic cost (AEC) would qualify as a bias (i.e., a manifestation detrimental to local optimality) rather than as the result of an appropriate inference.

Twenty-two healthy adults gave their consent to participate in this experiment and were tested one after the other. There was no a priori power calculation as we did not know what effect to expect, so this sample size (N = 22) was roughly deemed appropriate given previous experiments investigating similar phenomena.

## Choice

As illustrated in *Figure 2a*, participants tended to accept the high-effort option more often as the attached incentive level increased, and as the cost level decreased (with a sigmoid-like nonlinearity in the acceptance rate curve, characteristic of floor and ceiling effects for low and high differences between options, respectively). For a more quantitative yet straightforward assessment of these effects, we fitted a sigmoidal model of choice, including the REC and incentive levels attached to the HEHG option ($REC_{HEHG}$ and $Incentive_{HEHG}$) as explanatory variables. Choice data (but not rating data) from four participants were excluded before model fitting due to a total rate of 'no' (resp. 'yes' for one participant) choices ≤5/108, that is, a percentage of 'yes' (resp. 'nno') choices >95%. Unsurprisingly, we found a negative effect of $REC_{HEHG}$ ($\bar{\beta} = -1.1$, $CI = \left[-1.5; -.62\right]$, $t(17) = -5.0$, $p = 1.2 \cdot 10^{-4}$), and a strong positive effect of $Incentive_{HEHG}$ (*Figure 2a*, bottom, $\bar{\beta} = 4.1$, $CI = \left[3.1 ; 5.1\right]$, $t\left(17\right) = 8.8$, $p = 9.4 \cdot 10^{-8}$), on the probability to accept the HEHG offer.

## Anticipated energetic cost (AEC)

As expected, anticipated energetic cost (AEC) ratings about routes were strongly influenced by the REC of these routes ($\bar{\beta} = 15$, $CI = \left[14; 17\right]$, $t\left(21\right) = 20$, $p < 10^{-14}$), whose effect was tested in a generalized linear model (GLM) that included both REC and incentive levels as factors of interest, the latter as a proxy for incentive utility, which is typically a supralinear function of incentive magnitude (*Glimcher and Fehr, 2013*; *Schoemaker, 1982*). The effect of the REC was significant despite a 'regression-to-the-mean' bias (i.e., overestimation for low levels and underestimation for high levels, apparent on *Figure 2b*, top). Thus, after training, participants were able to estimate energetic cost of a running route with good accuracy (82% ± 3%), defined as the complement to distance from target (REC-AEC, see 'Materials and methods').

Most interestingly, we found a robust *positive* effect of incentive level (i.e., ordinal value) on energetic cost rating (*Figure 2b*, bottom, $\bar{\beta} = 2.0$, $CI = \left[1.2; 2.7\right]$, $t\left(21\right) = 5.3$, $p = 2.8 \cdot 10^{-5}$). That is, all other things being equal, participants tended to anticipate a higher energetic cost for better-rewarded running sequences. Because the relation between incentive level and energetic cost rating seemed in fact slightly supralinear (see *Figure 2b*), and in order to test more directly for the effect of

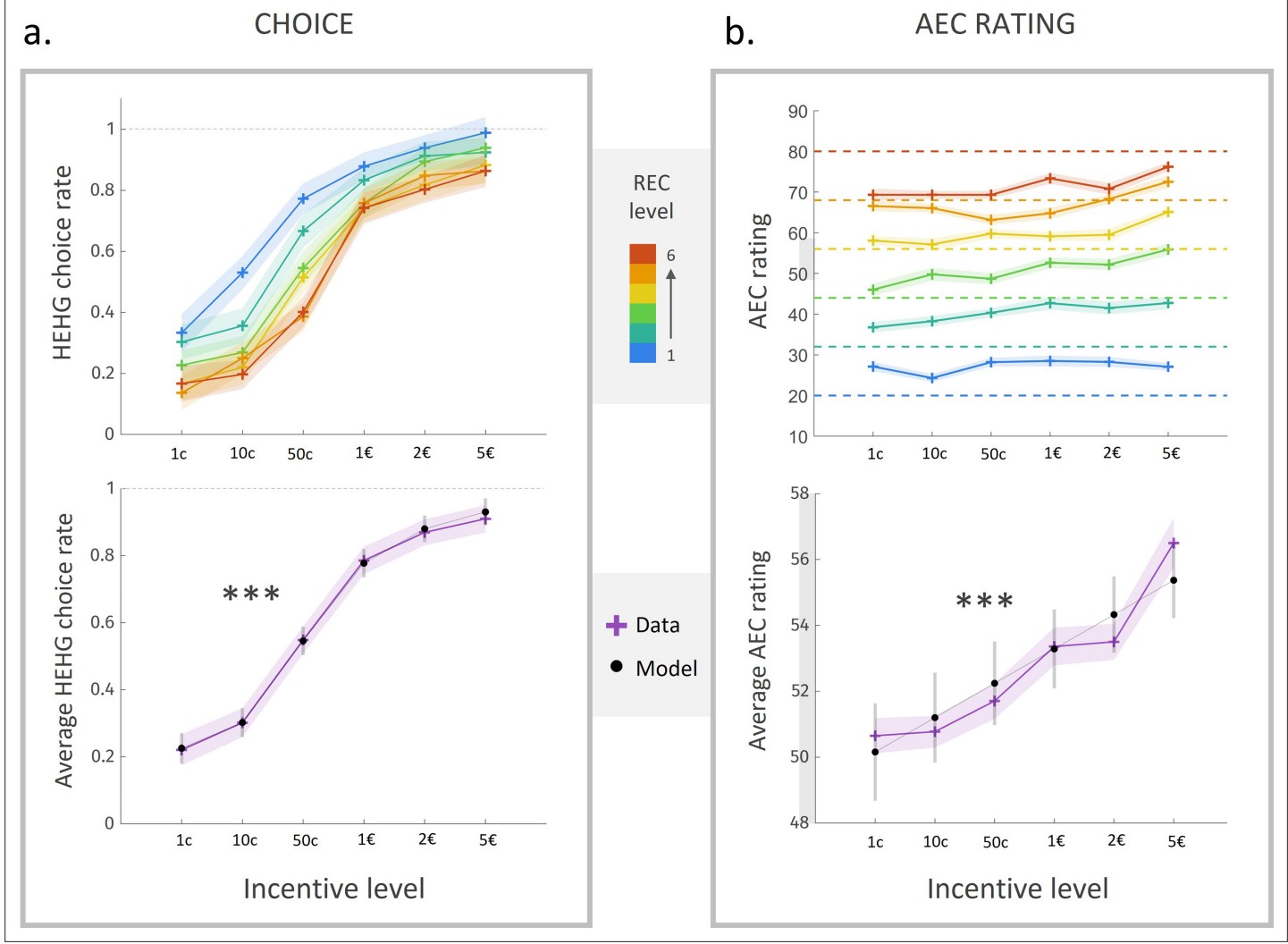

**Figure 2.** Choice rate and energetic cost rating in experiment 1. (**a**) Rate of acceptance of the high-effort high-gain (HEHG) offer, plotted against incentive level. Top: cross markers indicate acceptance rate averaged across all participants for a given incentive level (x-axis) and a given real energetic cost (REC) level (marker colour). Bottom: purple cross markers indicate acceptance rate, averaged across all REC levels and across all participants for a given incentive level. Black dots indicate averaged predictions of a logistic regression model (see main text). *** Positive effect of incentive level on acceptance rate, p<0.001. (**b**) Anticipated energetic cost (AEC) rating, plotted against incentive level. Top: cross markers indicate AEC ratings averaged across all participants for a given incentive level (x-axis) and a given REC level (marker colour). Dotted lines indicate the target rating for each REC level. Bottom: purple cross markers indicate AEC rating, averaged across all REC levels and across all participants for a given incentive level. Black dots indicate averaged predictions of a generalized linear model (GLM, see main text). ***Positive effect of incentive level on AEC rating, p<0.001. Solid lines represent a linear interpolation between averaged data points (or model predictions) for illustrative purposes only. Shaded area around curves or error bars across dots represent the standard error to the mean (s.e.m), computed across participants.

the incentive *magnitude* (i.e., cardinal value, which is the feature primarily relevant to participants), we fitted the following complementary model:

$$AEC = \beta_0 + \beta_1 \cdot REC + \beta_2 \cdot I_m^\alpha$$

where $I_m$ is the incentive magnitude and $\alpha$ a nonlinearity parameter that we expected to be below 1, in line with the typical observation of a concave magnitude-utility mapping (**Glimcher and Fehr, 2013**; **Schoemaker, 1982**).

We found a significant positive impact of the transformed incentive magnitude onto energetic cost rating ($\bar{\beta}_2 = .32$, $CI = [.18; .46]$, $t(21) = 4.7$, $p = 1.1 \cdot 10^{-4}$), with an average nonlinearity parameter $\bar{\alpha} = 0.74$.

Given the strong positive relation between incentive magnitude and acceptance rate of the high-effort option, one could hypothesize that the impact of incentive level on energetic cost rating was fully mediated by choice, with 'yes' choices leading to higher cost estimates than 'no' choices. However, the incentive effect on energetic cost rating remained strongly significant even when the *Incentive* predictor was orthogonalized to a binary *Choice* predictor $(\bar{\beta} = 1.9,\ CI = [1.1; 2.8],\ t(21) = 4.9,\ p = 8.0 \cdot 10^{-5})$. Moreover, alternative regressors $Incentive_{HEHG}$ and $Incentive_{LE0G}$ conditional on the choice made (respectively the HEHG option or the LE0G one) were both significant positive predictors of energetic cost rating $(HEHG:\ \bar{\beta} = 2.4,\ CI = [1.4; 3.4],\ t(20) = 5.0,\ p = 6.9 \cdot 10^{-5}; LE0G:\ \bar{\beta} = 2.0\ CI = [.70; 3.4],\ t(18) = 3.2,\ p = 5.0 \cdot 10^{-3};)$, meaning that the effect of *Incentive* on AEC was present within both 'yes' and 'no' trials. Therefore, prospective reward inflated AEC over and above choices made by the participants.

## Interim discussion and new hypotheses

In experiment 1, we observed an unexpected *positive* impact of potential reward onto anticipated effort cost. Because it was opposite to our predictions, a replication study was necessary to better establish this finding. We took this opportunity to specify the underlying cognitive mechanisms. Below are described the three types of cognitive mechanisms that we considered, hereafter denoted as the 'belief', 'affect', and 'decision' scenarios.

First, given that most healthy individuals hold the belief that obtaining more reward typically requires investing more effort (*Benabou and Tirole, 2006*; *Frieze and Snyder, 1980*; *Nalebuff and Stiglitz, 1983*; *Van-Yperen and Duda, 2007*), our participants might have used the incentive cue as implicit evidence informing their effort cost estimation. This hypothesis is in line with a notion of learned industriousness, according to which the association between effort and reward levels is conditioned through reinforcement mechanisms (*Eisenberger, 1992*; *Inzlicht et al., 2018*). One slightly different possibility is that participants may have explicitly assumed that reward and effort cost were correlated in this specific task. We henceforth refer to this account as the 'belief' scenario, which can be rephrased as individuals carrying a non-flat p(Cost|Incentive) representation, which they integrate as a likelihood signal into a posterior cost estimate. This belief scenario is the basis of ideal Bayesian observer models that capture the integration of reward magnitude into retrospective effort estimates, in paradigms where effort and reward levels are actually correlated by design (*Pooresmaeili et al., 2015*; *Rollwage et al., 2018*).

Second, because of its personal relevance, information about reward is likely to trigger an affective reaction from the viewer, as previously demonstrated in similar contexts (*Knutson and Greer, 2008*; *Wu et al., 2014*). In line with the affect-as-information theory, and its associated observations that internal affective states are incorporated as evidence into various kinds of judgements (*Clore and Huntsinger, 2007*; *Clore and Storbeck, 2006*; *Schwarz, 2012*; *Storbeck and Clore, 2008*), affect may have mediated the impact of incentives on cost anticipation. In the psychology literature, affective states are classically parametrized in a two-dimensional space: arousal by valence (*Posner et al., 2005*; *Russell, 1980*). The mediating role of affect could therefore be specific to one of these dimensions, or rely on both. In particular, since both prospective and exerted effort has been shown to increase arousal (*Gellatly and Meyer, 1992*; *Schmidt et al., 2009*; *Varazzani et al., 2015*; *Vassena et al., 2014*), it is possible that the cognitive system responsible for effort anticipation may use arousal state as one of its inputs. Moreover, because prospective rewards also increase arousal (*Knutson and Greer, 2008*; *Wu et al., 2014*), individuals might misinterpret incentive-triggered arousal as a cue about the upcoming effort, such that AEC would be higher when a higher reward is at stake. A similar reasoning could apply in a valence-based scenario since it is well established that prospective reward positively modulates affective valence (*Knutson and Greer, 2008*). Yet this scenario would imply that prospective effort is assigned a positive valence in our group of participants, which is not implausible given that they all practised sports regularly (this was one of our inclusion criteria). We henceforth refer to this as the 'affect' scenario, which comprises an 'arousal' and a 'valence' variant.

Third, at a motivational level, information about reward is a crucial input to the cost–benefit computations that adjust both the direction and intensity of behaviour, according to standard decision theory (*Croxson et al., 2009*; *Glimcher and Fehr, 2013*; *Rangel et al., 2008*). Since our participants were systematically required to decide whether to accept a new reward–effort combination, it is likely that they injected the reward information into some expected value computation which would later guide

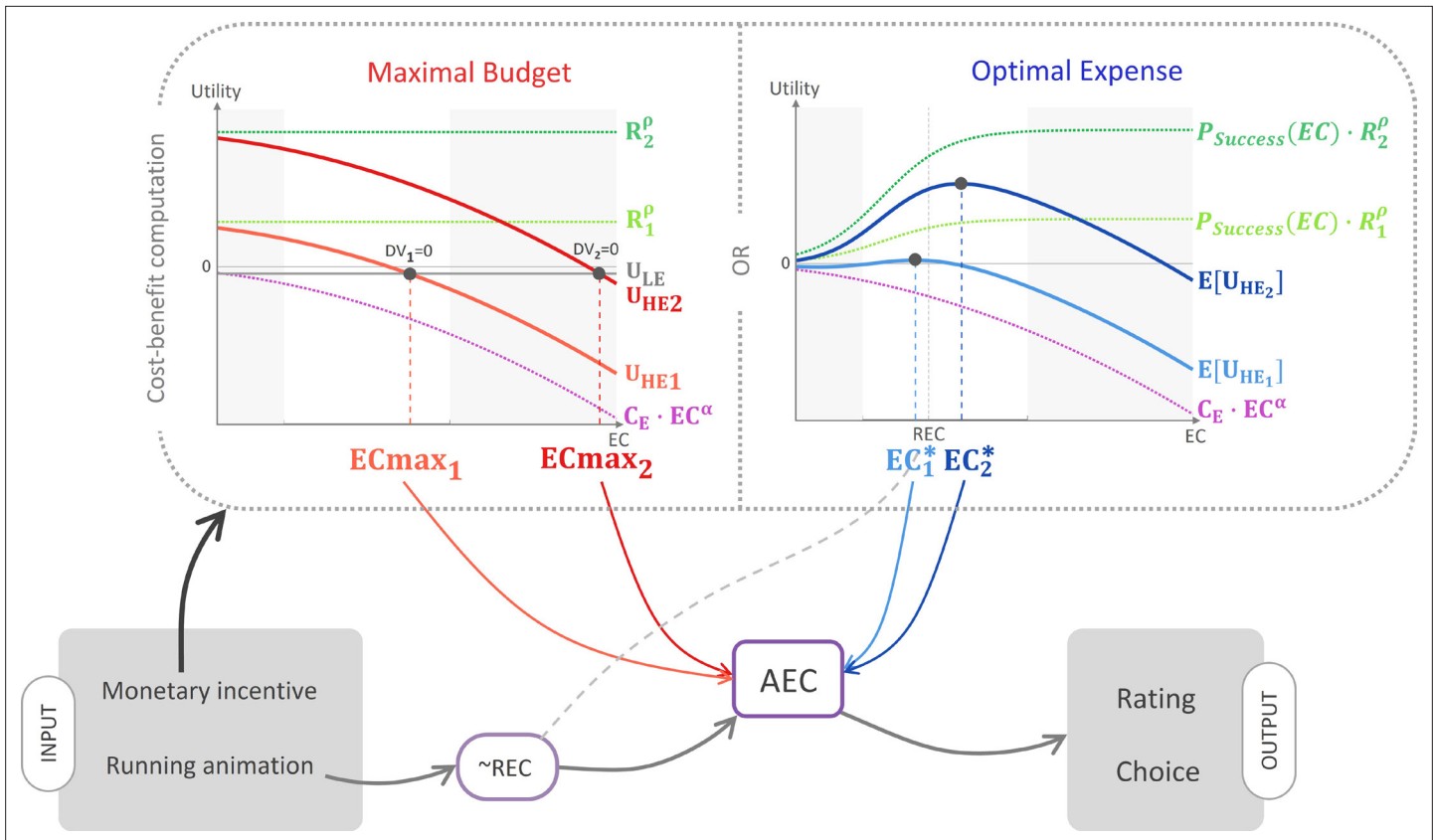

**Figure 3.** Schematic illustration of the 'cost–benefit' scenarios. According to this hypothetical mechanism, information about the monetary incentive attached to the high-effort (HE) option enters a cost–benefit computation. According to the maximal budget variant (left panel), participants compute the maximal energetic cost above which the low-effort (LE) option would bear a higher utility (see *Equations 1–3* in main text). According to the optimal expense variant, they compute the energetic cost optimizing the expected utility attached to the HE option (see *Equations 7–9* in main text). In both panels, the computation is illustrated for two different reward magnitudes, R1 (small, light colour) and R2 (large, dark colour). Light grey areas correspond to domains outside the energetic cost rating scale. Both variants predict that larger incentives would result in a higher anticipated energetic cost (AEC) computation (see *Equations 4 and 10*), which also integrates a pre-estimate that is exclusively based on the running animation depicting the multi-segment route, and is thus a noisy reflection of the real energetic cost (REC). Finally, the resulting AEC estimate affects both energetic cost rating and choice between the HE offer and the LE reference option (see *Equations 5 and 6* and *Equations 11 and 12*).

their choice. From there, two specific mechanisms can account for a positive influence of prospective reward on anticipated effort cost. In any case, the most parsimonious formalization is that some output of the cost–benefit computation (dependent on prospective reward) would be integrated as evidence about the energetic cost. Our two decision models detailed below differ as to the nature of the computational output that is integrated into cost estimation.

According to what we hereafter call the 'maximal budget' hypothesis, as soon as individuals receive complete information about the potential reward, they compute the maximal energetic cost they would be willing to invest, that is, the cost above which they would rather choose the low-effort option (see illustration in *Figure 3*, left). This preliminary computation may entail easier subsequent decision-making since choice could then result from a direct comparison between the maximal budget and the estimated energetic cost. For example, when offered a X€ reward, the participant might pre-commit to a Y effort cost (in intuitive terms: "For this reward, I'm ready to run this kind of route").

Building on standard decision theory, we modelled choice as a logistic transformation of a decision variable DV that integrates the reward levels R and the effort cost estimates EC associated to the two options. Here, we assume a subtractive parabolic effort–cost discounting of subjective incentive values (*Białaszek et al., 2017*; *Hartmann et al., 2013*), such that the decision variable, that is, the difference in utility U between the high-effort (HE) option and the low-effort (LE) one, can be written as

$$DV = U_{HE} - U_{LE} = \left(R^\rho - C_E \cdot EC^\alpha\right) - \left(0 - C_E \cdot EC_0^\alpha\right) \tag{1}$$

where $R$ is the reward obtained upon completion of the high-effort route, $\rho$ is the nonlinearity parameter controlling the curvature of magnitude-to-utility mapping (typically, $\rho < 1$; **Glimcher and Fehr, 2013**; **Schoemaker, 1982**), $C_E$ is the scaling coefficient of the effort cost discounting variable ($EC$ and $EC_0$ corresponding to the energetic cost of the high-effort and low-effort options, respectively), and $\alpha$ is the nonlinearity parameter controlling the curvature of effort discounting.
Thus:

$$DV = R^\rho - C_E \cdot \left(EC^\alpha - EC_0^\alpha\right) \tag{2}$$

therefore, the maximal energetic cost such that DV = 0 is

$$EC_{max} = \left(EC_0^\alpha + \frac{R^\rho}{C_E}\right)^{\frac{1}{\alpha}} \tag{3}$$

The main proposal of this 'maximal budget' model is that participants' AEC derives from a mixture between the relevant visual information (slope and speed, underlying the REC) and the maximal energetic cost they are willing to expend:

$$AEC = B_0 + B_1 \cdot (\gamma \cdot REC + (1 - \gamma) \cdot EC_{max}) \tag{4}$$

where $\gamma \in 0, 1 \lbrack$ is a linear parameter adjusting the weight of REC, and $B_0$ and $B_1$ are intercept and slope parameters controlling an affine transformation from the REC-$EC_{max}$ mixture to AEC rating (in order to descriptively account for the regression to the mean observed in the data). As this biased cost judgement likely reflects the cost which participants are planning to invest, it may in turn be incorporated in an updated decision variable:

$$DV' = R^\rho - C_E \cdot \left(AEC^\alpha - EC_0^\alpha\right) \tag{5}$$

This updated decision variable then leads to a choice via a logistic transformation:

$$P\left(Choice = Yes\right) = sigmo\left(\beta_0 + \beta_1 \cdot DV'\right) \tag{6}$$

where $sigmo\left(x\right) = \frac{1}{1+e^{-x}}$ and $\beta_0$ and $\beta_1$ correspond respectively to bias and inverse temperature parameters. As evidenced by **Equations 3 and 4**, this model indeed predicts a positive impact of monetary incentives on AEC.

In an alternative scenario, which we call the 'optimal expense' hypothesis, we propose that it is the output of an *optimal expense* computation, instead of a *maximal budget* one, that is subsumed as evidence about the actual cost (see illustration in **Figure 3**, right). Here, we build upon the intuition that participants might not fully internalize the mapping rule between expended energy and the probability of reaching their goal (obtaining the reward). In our experiment, the effort–reward contingency is deterministic: the participant should be sure to win the full reward if running the high-effort route, and nothing otherwise. Thus, the mapping from effort to reward obtainment probability is a step function with a 0–1 discontinuity when the REC is reached. Indeed, as this discontinuity is due to the fact that energetic expense is strongly constrained in our experimental setting (running on a treadmill at a defined slope and speed), it might not reflect most ecological situations, in which increasing effort would naturally increase the probability of reaching one's goal in a continuous fashion. Following on this principle, the participant might think that exerting a bit more effort than required would help secure the outcome (in intuitive terms: "For this reward, I should make some extra effort to maximize my chances to get it in the end").

Therefore, we propose that individuals hold a sigmoidal (rather than step-like) $P\left(Success|\, Energetic\; expense\right) = f\left(Energetic\; expense\right)$ internal mapping function, whose inflection point abscissa and slope may be adjusted depending on the specifics of the considered task. Here, to make the sigmoid parameter values easier to interpret, we express the energetic expense variable as relative to the REC and normalized by the maximal possible cost $E_{max}$ :

$$P\left(Success|EC\right) = sigmo\left(\kappa_0 + \kappa_1 \cdot \frac{EC - REC}{E_{max}}\right) \tag{7}$$

where $P\left(Success|EC\right)$ is the subjective probability of succeeding at completing the route given the expended energetic cost $EC$, $\kappa_0$ is a bias parameter, and $\kappa_1$ is an inverse temperature parameter. In line with classical expected utility theory (**Friedman and Savage, 1952**), this subjective probability of success enters the computation of the decision variable by scaling the utility of prospective outcomes, which is discounted by expected cost:

$$E\left[U_{HE}\left(EC\right)\right] = P\left(Success|EC\right) \cdot R^\rho - C_E \cdot EC^\alpha \tag{8}$$

By definition, the optimal energetic cost to expend is the one maximizing $E\left[U_{HE}\left(EC\right)\right]$:

$$EC^* = \underset{EC}{argmax}\left(E\left[U_{HE}\left(EC\right)\right]\right) \tag{9}$$

As with the 'maximal budget' hypothesis, we postulate that participants incorporate this optimal energetic cost (which they intend to pay) as a piece of evidence about the actual energetic cost of the route:

$$AEC = B_0 + B_1 \cdot \left(\gamma \cdot REC + \left(1 - \gamma\right) \cdot EC^*\right) \tag{10}$$

where parameters $B_0$, $B_1$, and $\gamma$ have the same meaning as in **Equation 4**. The decision variable being the difference between $E\left[U_{HE}\left(AEC\right)\right]$ and $U_{LE}$, it is written as

$$DV' = P\left(Success|AEC\right) \cdot R^\rho - C_E \cdot \left(AEC^\alpha - EC_0^\alpha\right) \tag{11}$$

Finally, this decision variable is incorporated into a softmax rule to determine choice probability:

$$P\left(Choice = Yes\right) = sigmo\left(\beta_0 + \beta_1 \cdot DV'\right) \tag{12}$$

We henceforth refer to this couple of models as the 'decision' scenario, which comprises a 'maximal budget' and an 'optimal expense' variant.

## Practical implementation and predictions

In order to assess the validity of these three alternative — albeit not mutually exclusive — scenarios, we formulated critical modifications of our task design in preparation for a second experiment (see **Figure 4**). These changes were devised in such a way that each of our hypotheses would make a specific set of predictions about measured behavioural outputs.

Under the 'belief' scenario, we reasoned that if the orthogonality between monetary incentive and effort cost was made clear to participants, we should lose the significant impact of prospective reward on AEC. We therefore explained to participants that effort and reward levels associated to running routes were independently and randomly selected. We also made this orthogonality intuitive and salient by introducing a lottery animation at the beginning of each trial. This virtual lottery depicted the roll of a slot machine, where a running route was (randomly) selected in the left slot and a monetary amount was selected (independently) in the right slot. Additionally, the 'belief' hypothesis predicted that the extent to which prospective reward impacted AEC should correlate, across individuals, with the strength of the general belief in a reward–effort positive correlation. We measured the strength of this belief by appending a computerized question after arousal rating, which was phrased as follows: "Do you think that, in life, the less effort one makes, the less reward one receives?", with four possible answers, ranging from 'absolutely yes' to 'not at all'.

Under the 'affect' scenario, incidental fluctuations in affective state should have repercussions on AEC. Thus, we introduced two novel features to the task in order to manipulate affective arousal and valence independently. First, to address the 'valence' variant, we complemented prospective gains with prospective losses (in a distinct set of trials), which, we assumed, would trigger negative affect (proportionally to loss magnitude), which should in turn decrease AEC. In loss trials, choices were symmetrical to those implemented in gain trials: the high-effort offer was associated with losing nothing, whereas the reference low-effort option was associated with a variable amount of money to be lost. Second, in order to address the 'arousal' variant, we added a musical background to two-thirds of the running animations. Based on previous reports of music-based arousal manipulation (**Gingras et al., 2015**; **Gomez and Danuser, 2007**), we expected our musical extracts to modulate

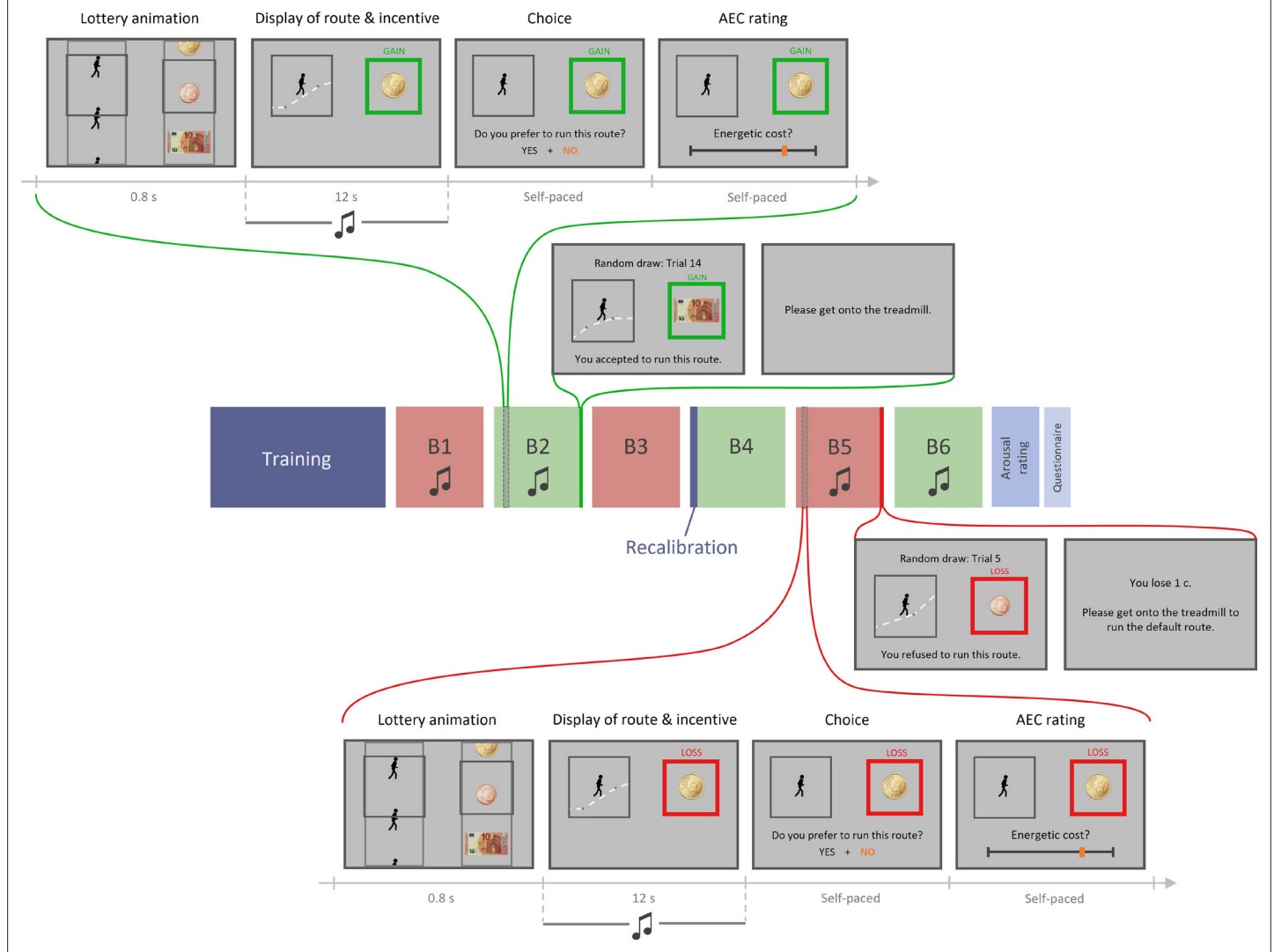

**Figure 4.** Experiment 2 design. Top part, top row: example trial in the gain condition with music. Each trial includes four phases: (1) lottery animation, during which a running route and an incentive level are independently and (seemingly) randomly selected; (2) display of the running route animation and the associated incentive (potential gain here) while a musical extract plays in participant's headset; (3) choice between this high-effort high-gain (HEHG) offer and an implicit low-effort 0-gain (LE0G) option; and (4) rating of anticipated energetic cost (AEC). Top part, bottom row: random draw performed at the end of a gain block among its 18 just-completed trials. Choice made by the participant is reminded then executed: in the gain condition, if the participant accepted the HEHG offer, she must get onto the treadmill and complete the effortful running route, after which the associated gain is added to her total earnings. If she declined the high-effort offer, she must complete a default low-effort running route, and her total earnings remain unchanged. Middle part: structure of a session. Each session includes a training phase (with practice on the treadmill and calibration of energetic cost rating with objective feedback), three blocks of 18 task trials, a recalibration (AEC ratings followed by objective feedbacks), three additional task trial blocks, a rating of arousal induced by all 72 musical extracts, and eventually a question about their belief in a general effort–reward correlation. Here, gain and loss conditions are alternated across blocks (green and red, respectively), and all blocks except B3 and B4 include musical extracts during display of route and incentive. Bottom part, bottom row: example trial in the loss condition with music. It is equivalent to the gain condition shown in top part, except that participants choose between a high-effort 0-loss (HE0L) and a low-effort high-loss condition (LEHL). Bottom part, top row: random draw performed at the end of a loss block among its 18 just-completed trials. Choice made by the participant is reminded then executed: in the loss condition, if the participant accepted the HE0L offer, she must get onto the treadmill and complete the effortful running route, and her total earnings remain unchanged. If she declined the high-effort offer, she must complete a default low-effort running route, after which the associated loss is subtracted from her total earnings.

participants' arousal, depending on the varying low- or high-level features of each extract (slow versus fast tempo, high versus low predictability, soothing versus epic impression, etc.). As arousal induction may vary a lot across individuals, we played again each music extract at the end of the experiment, and had every participant reporting subjective arousal on a rating scale. Using eye-tracking to

measure pupil diameter during the rating task, we checked that subjective report of arousal level was following, across music extracts, arousal level in a physiological sense (activation of the autonomic nervous system). Additionally, regardless of the characteristics of our musical extracts, we expected the presence of music to increase average arousal, compared to the testing blocks in which music would be absent (*Gomez and Danuser, 2004*). Therefore, this manipulation provided us with two ways (using subjective rating and pupil dilation) of testing for a positive impact of incidental arousal fluctuations on AEC, which was predicted by the arousal variant of the affect scenario.

Finally, under the decision scenario, and given the previously introduced experimental changes, we expected a distinct set of observations: a positive effect of incentive on AEC in both gain and loss conditions (and despite the introduction of a lottery animation), no inter-individual correlation between the magnitude of this effect and the strength of the belief in a reward–effort correlation, and finally, no effect of our music-based arousal manipulation on AEC. To further arbitrate between the 'maximal budget' and 'optimal expense' variants, we planned to perform group-level Bayesian model selection based on choice and rating data.

## Results: Experiment 2

Twenty-four healthy adults gave their consent to participate in this experiment and were tested one after the other. Given the mean and standard deviation of the main effect size in experiment 1 (effect of gain level on energetic cost rating), this sample (N = 24) was sufficient to get a detection power of 99% with significance threshold set at 0.05.

## Choice

Here, we replicated the observations made in experiment 1, with an extension into the loss domain: participants tended to accept the high-effort (HE) option more often as gain or loss level increased, and as REC level decreased. We fitted a sigmoidal model of choice between high-effort and low-effort options, including gain and loss magnitudes $Gain_{HEHG}$ and $Loss_{LEHL}$, as well as $REC_{HE}$, as explanatory variables. Choice data (but not rating data) from 10 participants were excluded before model fitting due to a total rate of 'no' choices ≤5/108, thus a percentage of 'yes' choices >95%. As expected, we found a strong positive effect of $Gain_{HEHG}$ (*Figure 5a*, bottom, $\bar{\beta} = 3.7$, $CI = [2.8 ; 4.6]$, $t(13) = 8.5$, $p = 1.1 \cdot 10^{-6}$) and $Loss_{LEHL}$ ($\bar{\beta} = 3.7$, $CI = [2.7 ; 4.8]$, $t(13) = 7.8$, $p = 2.9 \cdot 10^{-6}$), and a negative effect of $REC_{HE}$ ($\bar{\beta} = -1.2$, $CI = [-1.7; -.76]$, $t(13) = -5.6$, $p = 8.5 \cdot 10^{-5}$) on the probability to prefer the high-effort offer over the low-effort one.

## Anticipated energetic cost

In order to evaluate the impact of experimental factors on anticipated effort cost (AEC) rating, and similarly to experiment 1, we fitted a GLM comprising the following regressors: the REC, the prospective gain level (0 in loss condition), and the prospective loss level (0 in gain condition) (*Figure 5b*). Beyond the unsurprising positive effect of REC on AEC ($\bar{\beta} = 15$, $CI = [14 ; 16]$, $t(23) = 24$, $p < 10^{-14}$), we replicated the positive effect of prospective gain level ($\bar{\beta} = 2.8$, $CI = [1.4 ; 4.3]$, $t(23) = 4.0$, $p = 6.3 \cdot 10^{-4}$). The effect size was even larger than in experiment 1, but this difference was not significant ($\bar{\beta} = -.89$, $CI = [-2.6 ; .78]$, $t(44) = -1.1$, $p = .29$). The significant impact of gain level, despite the demonstration that reward and effort were independent, thus contradicts one of the predictions of the 'belief' scenario. Overall, when pooling participants across both experiments (N = 46), we found a very robust positive effect of prospective gain level on AEC rating ($\bar{\beta} = 2.4$, $CI = [1.6 ; 3.2]$, $t(45) = 5.8$, $p = 5.7 \cdot 10^{-7}$). Importantly, we also found a significant positive effect of prospective loss level on AEC ($\bar{\beta} = 2.8$, $CI = [1.2 ; 4.4]$, $t(23) = 3.6$, $p = .0016$), which contradicts the main prediction from the valence variant of the 'affect' scenario. These effects were also significant in a model of AEC depending on gain and loss magnitudes rather than levels (gain: $\bar{\beta} = 1.6$, $CI = [.72 ; 2.5]$, $t(23) = 3.7$, $p = .0012$; loss: $\bar{\beta} = 1.7$, $CI = [.66 ; 2.8]$, $t(23) = 3.4$, $p = .0027$).

Additionally, as in experiment 1, the effects of both gain and loss levels on AEC remained strongly significant when the *Gain* and *Loss* predictors were orthogonalized to a binary *Choice* predictor (*Gain*: $\bar{\beta} = 0.87$, $CI = [0.29; 1.5]$, $t(23) = 3.1$, $p = 5.3 \cdot 10^{-3}$; *Loss*: $\bar{\beta} = 2.2$, $CI = [1.0; 3.3]$, $t(23) = 3.9$, $p = 6.9 \cdot 10^{-4}$). Moreover, alternative regressors $Gain_{HEHG}$ and $Gain_{LE0G}$ conditional on the choice made (respectively the HEHG option or the LE0G one in gain trials) were both significant positive predictors of AEC rating (*HEHG*: $\bar{\beta} = 3.9$, $CI = [1.8; 6.1]$, $t(23) = 3.9$, $p = 7.7 \cdot 10^{-4}$;

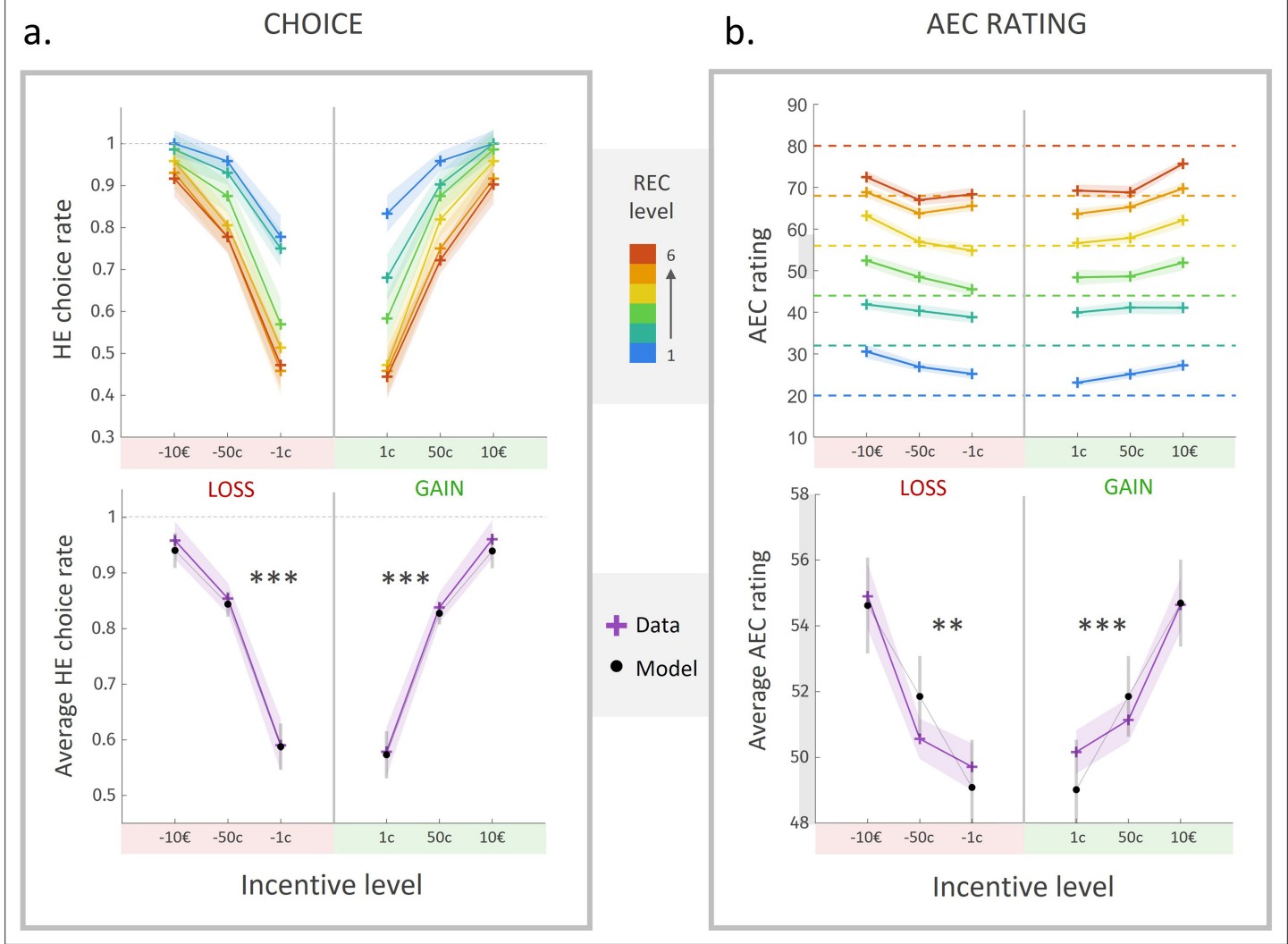

**Figure 5.** Choice rate and effort cost rating in experiment 2. (**a**) Rate of acceptance of the high-effort offer, plotted against incentive level. Top: cross markers indicate acceptance rates averaged across all participants for a given incentive level (x-axis) and a given real energetic cost (REC) level (marker colour). Bottom: cross markers indicate acceptance rate, averaged across all REC levels and across all participants for a given incentive level. Black dots indicate averaged predictions of a logistic regression model (see main text). ***Positive effect of loss and gain level on acceptance rate, both p<0.001. (**b**) Anticipated energetic cost (AEC) rating, plotted against incentive level. Top: cross markers indicate AEC ratings averaged across all participants for a given incentive level (x-axis) and a given REC level (marker colour). Dotted lines indicate the target rating for each REC level. Bottom: cross markers indicate AEC ratings, averaged across all REC levels and across all participants for a given incentive level. Black dots indicate averaged predictions of a generalized linear model (GLM, see main text). **Positive effect of loss level on AEC rating, p<0.005. ***Positive effect of gain level on AEC rating, p<0.001. Solid lines represent a linear interpolation between averaged data points for illustrative purposes only. Shaded area around each curve represents the standard error to the mean (s.e.m), computed across participants.

$LE0G$: $\bar{\beta} = 4.8$ $CI = [1.9; 7.8]$, $t(10) = 3.7$, $p = 4.5 \cdot 10^{-3}$), as were the regressors $Loss_{HE0L}$ and $Loss_{LEHL}$ ($HE0L$: $\bar{\beta} = 3.8$, $CI = [1.5; 6.1]$, $t(23) = 3.5$, $p = 2.0 \cdot 10^{-3}$; $LEHL$: $\bar{\beta} = 4.9$ $CI = [2.0; 7.8]$, $t(10) = 3.7$, $p = 3.9 \cdot 10^{-3}$), meaning that the effect of both prospective gain and loss on AEC was present within both 'yes' and 'no' trials. Therefore, prospective gains and losses inflated anticipated energetic cost over and above choices made by the participants.

## Pupil dilation and arousal manipulation

Next, we assessed the impact of our arousal manipulation procedure on AEC rating. First, we checked that arousal ratings made about each musical extract were actually reflecting their arousal response to musical extracts. We found that individual arousal ratings for a given musical extract were significantly positively predicted by arousal ratings averaged across all other participants

$(\overline{\beta} = .56, CI = [.46; .72], t(23) = 9.2, p = 3.9 \cdot 10^{-9}$; Figure S1). This finding suggests that ratings were not given randomly but instead were made to reflect participants' experienced arousal. We also observed that arousal rating was a significant positive predictor of pupil diameter around 2 s after music onset (time window: [1.67–2.55] s post music onset, $p = 3 \cdot 10^{-4}$, corrected for multiple comparisons; Figure S2), suggesting that ratings faithfully reflected the arousal state actually felt by participants while listening to the musical extracts. Second, we incorporated these individual arousal ratings as an additional linear regressor in our model of AEC rating (see above). We found no significant effect of this predictor ($\overline{\beta} = .19, CI = [-.35 ; .74], t(23) = .73, p = .47$), nor did we find an effect of a binary 'music/nomusic' regressor indicating whether music was played during a given trial ($\overline{\beta} = -.16, CI = [-.93 ; .61], t(23) = -.43, p = .67$). These observations suggest that incidental arousal fluctuations do not affect anticipated energetic cost, and therefore challenge the main prediction of the arousal variant of the 'affect' scenario.

### Belief in an effort => reward implication

We measured the individual strength of belief in an effort–reward correlation by asking the following question (via the computer), once all trials were completed: "Do you believe that in life, the less effort one makes, the less reward one gets?". Participants answered on a 4-point Likert scale ranging from "Not at all" to "Absolutely" (average reported belief strength = 2.9, standard deviation = 0.80). We found no correlation between the strength of their belief and the magnitude of the effect of gain or loss, or both gain and loss combined, on anticipated effort cost rating (resp. $r = -.18, p = .39; r = -.049, p = .82; r = -.12, p = .59$). This finding is in contradiction with one of the predictions of the 'belief' scenario.

### Exploring the decision scenario: Computational modelling

As introduced in the interim discussion, we propose two computational models which could explain the positive effect of goal value on anticipated energetic cost from high-level, decision-related cognitive processes. Specifically, the 'maximum budget' hypothesis proposes that, as soon as the goal value is known, individuals compute the corresponding maximal effort they are willing to invest to attain it. Subsequently, they (partly) intermingle what they are *willing to* invest with what they *must* invest. As an alternative, the 'optimal expense' scenario proposes that individuals compute the optimal amount of effort they should invest to maximize the expected utility of their action, that is, to secure the outcome, assuming the effort–outcome contingency is uncertain. This 'optimal effort' variable is then (partly) confused with the objective effort which is demanded from them.

In order to assess the significance of these hypotheses, and to arbitrate between them based on the relevant data (AEC ratings and yes/no choices), we fitted each of these two models (both accounting for rating and choice simultaneously) to the datasets from experiments 1 and 2. Within each model family, two variants were inverted. First was a 'standard' variant described in the interim discussion (with an additional loss term when fitting data from experiment 2, see 'Materials and methods'), in which the probability of choosing the high-effort option is a sigmoidal function of the AEC estimate (also derived from the model; see (*Equations 4 and 6*) in interim discussion and (*Equation 23*) in 'Materials and methods' for the 'maximal budget' model, and (*Equations 10 and 12*) in interim discussion and (*Equation 25*) in 'Materials and mmethods' for the 'optimal expense' model). Second was a 'null choice' variant in which choices did not depend on AEC estimates, but directly on the REC (i.e., the subjective distortions of effort cost were not incorporated into decision-making). This variant was obtained simply by replacing AEC with REC in (*Equation 23*) and (*Equation 25*).

Additionally, we fitted purely descriptive models featuring linear terms which accounted for the influence of prospective gain or loss on AEC rating. The first variant of these descriptive models – hereafter labelled 'linear' – only included main effects:

$$AEC = \beta_0 + \beta_1 \cdot REC + \beta_2 \cdot R + \beta_3 \cdot L$$

where $\beta_i$ are the linear coefficients of the independent variables, while the second one, 'Linear$_{\text{Interac}}$', further comprised terms of interaction with REC:

$$AEC = \beta_0 + \beta_1 \cdot REC + \beta_2 \cdot R + \beta_3 \cdot L + \beta_4 \cdot REC \cdot G + \beta_5 \cdot REC \cdot L$$

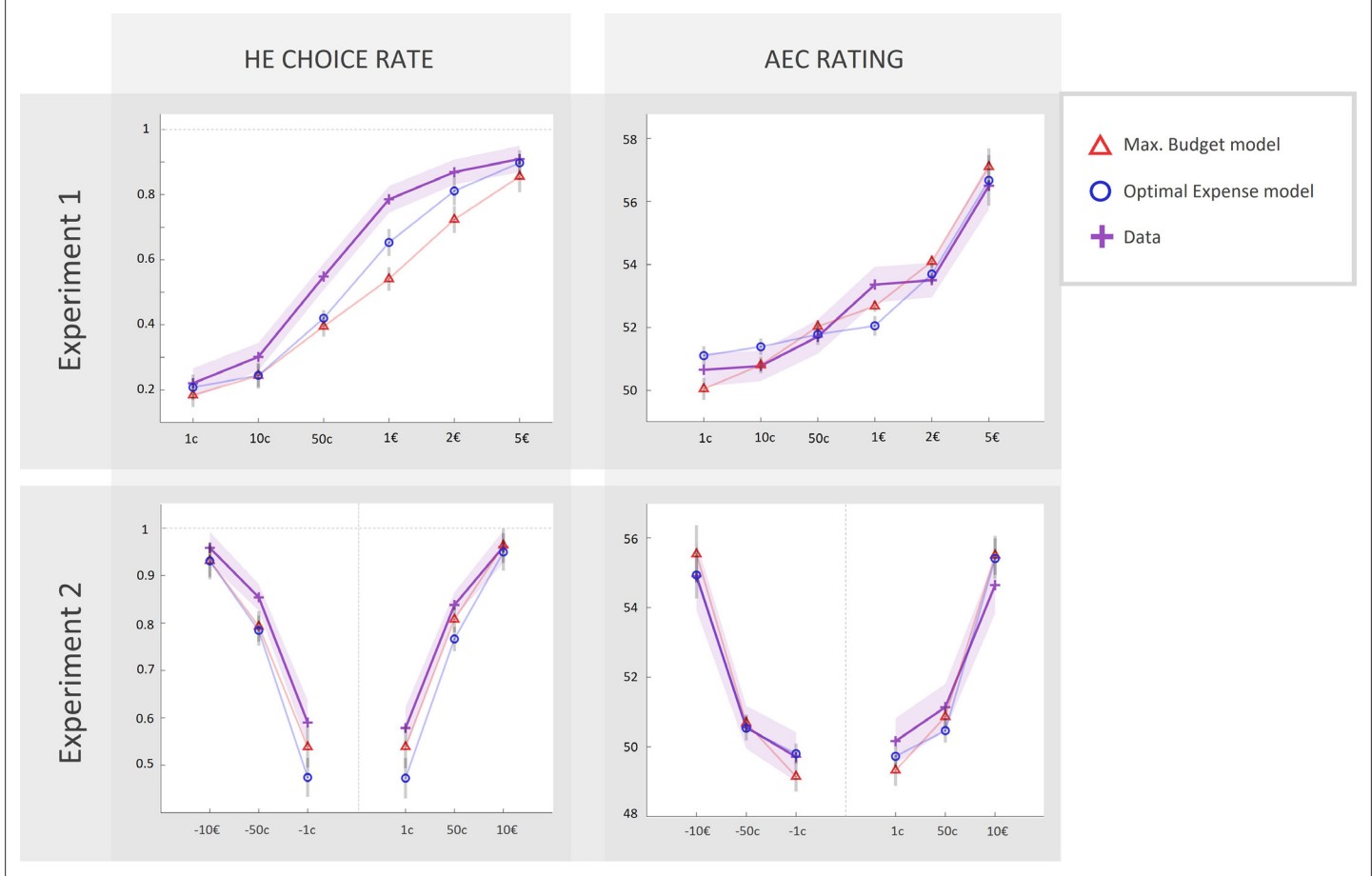

**Figure 6.** Fits of computational models to observed data in experiments 1 and 2 (N = 46). Left column: rate of acceptance of the high-effort offer, plotted against incentive levels. Right column: anticipated effort cost (AEC) rating, plotted against incentive level. Cross markers indicate observed responses averaged across all real energetic cost (REC) levels and across all participants in experiment 1 or 2 (top or bottom line, respectively) for a given incentive level. Triangles and circles indicate averaged responses generated by the 'maximal budget' and 'optimal expense' models, respectively, with parameters fitted to individual data (both ratings and choices in both experiments). Solid lines represent a linear interpolation between averaged observed or modelled responses for illustrative purposes only. Shaded area around each line represents the standard error to the mean (s.e.m), computed across participants.

We used the same equations to account for choice-making under these descriptive models as with the 'maximal budget' and 'optimal expense' models, either under the 'standard' or 'null choice' variant.

We first fitted each of our models to data from experiment 1 (with agnostic priors on parameters' density functions), which allowed us to derive empirically informed prior parameter distributions, before fitting the models again to data from experiment 2. Corresponding model fits for standard 'maximal budget' and 'optimal expense' models are illustrated in *Figure 6*. Both models do predict the qualitative pattern observed in high-effort choice rate and AEC rating.

In order to arbitrate between model families from a quantitative standpoint, we pooled free-energy approximations of model evidence across the two groups of participants (experiments 1 and 2) and performed several group-level random-effects Bayesian model family comparisons. First, we compared 'standard' models, wherein choice probability is a function of the model's AEC estimate, to 'null choice' models, wherein choice probability is a function of REC (see *Figure 7*, right part, top row). The protected exceedance probability ($PEP > 0.99$) strongly favoured the 'standard' family, providing support for the fact that AEC estimate, which incorporated several distortions of the REC, including those relating to gain or loss influences, was a better predictor of choice than REC itself. Second, we compared a 'cost–benefit' family (comprising both variants of 'maximal budget' or 'optimal expense' models) to a 'linear' family (comprising both variants of 'Linear' and 'Linear$_{Interac}$' models). This revealed a very likely dominance of the 'cost–benefit' family (again, $PEP > 0.99$), suggesting that most

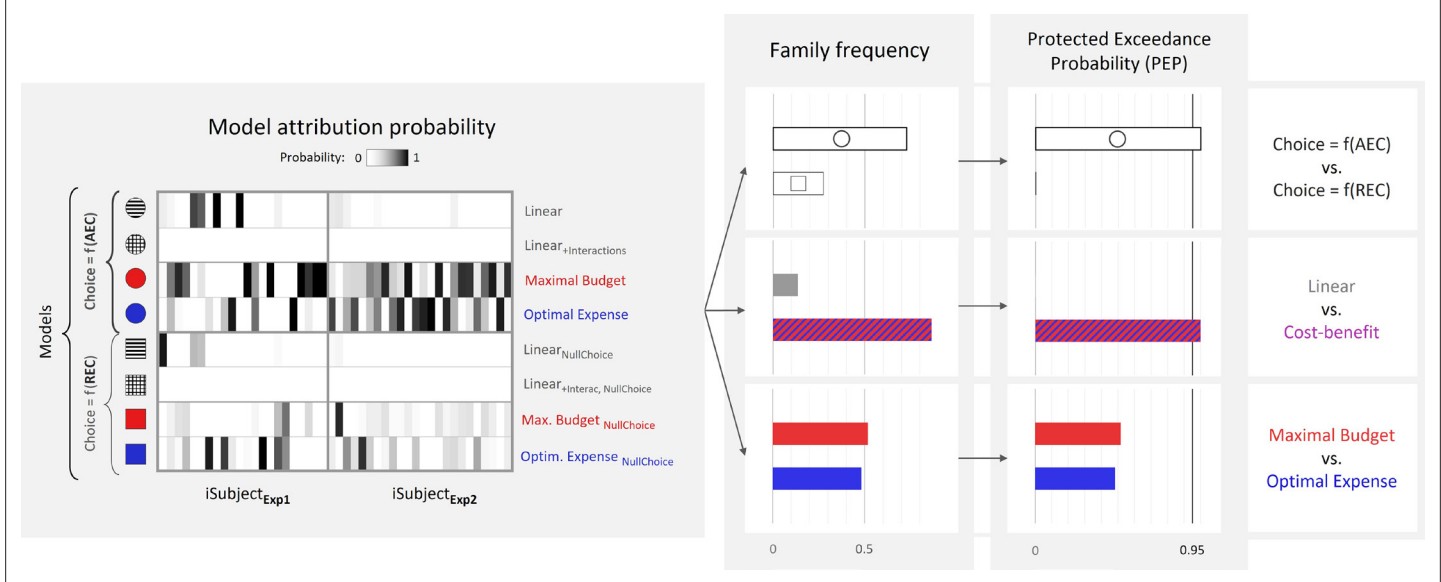

**Figure 7.** Bayesian model comparison across experiments 1 and 2 (N = 46). Left: model attribution probability for each participant in experiments 1 and 2 (x-axis) and each model variant (y-axis). First four models from top correspond to the 'standard' variant, wherein the probability of choosing the high-effort option depends on the anticipated effort cost (AEC), while last four models correspond to the 'null choice' variant, wherein this probability depends on the real energetic cost (REC). Right: estimated frequency and protected exceedance probability (PEP) of each model family. PEP is the probability that a given model family is more frequent than any other one in the population, corrected for chance fluctuations of observed individual model evidences. Here, $P\left(freq_{Choice=f(AEC)} > freq_{Choice=f(REC)}\right) > 0.99$, $P\left(freq_{Cost-benefit} > freq_{Linear}\right) > 0.99$ and $P\left(freq_{MaxBudget} > freq_{OptimExpense}\right) = 0.52$.

participants are indeed incorporating the output of a cost–benefit computation into their prospective cost judgement. However, a third comparison between 'maximal budget' and 'optimal expense' families did not reveal any likely dominance between the two ($PEP = 0.52$).

## Discussion

In this study, we investigated the putative influence of goal value onto the effort cost which individuals anticipate they will have to expend in order to achieve this goal. To this end, we devised a novel experimental paradigm in which healthy participants made prospective energetic cost estimates about composite running routes paired with a monetary outcome, whose magnitude was varied orthogonally to the actual energetic cost of the routes. In our first experiment, despite a financial incentive for accurate cost estimates, and contrary to our expectations, we found a *positive* impact of incentive magnitude on prospective effort cost estimates. To our knowledge, this is the first time such an effect has been isolated in a controlled laboratory setting. It questions a fundamental axiom of rational decision theory, according to which effort cost and goal value should be estimated independently of each other.

We next set out to elucidate the cognitive processes underlying the value dependency of cost estimates, which we hereafter call the decision bias. Three main cognitive scenarios could be put forth: (1) a 'belief' scenario, in which individuals incorporate information about goal value as a cue about effort cost, due to an implicit or explicit belief in a positive correlation between these two dimensions, (2) an 'affect' scenario in which affective states are modulated (along their valence or arousal dimension) by prospective gains or losses and in turn inform prospective effort cost judgements, and (3) a 'decision' scenario in which prospective effort cost judgements are partly biased by the output of cost–benefit evaluations related to effort investment (either the maximal effort one is willing to invest in the pursuit of the goal or the optimal effort one should invest to maximize their expected utility). We extended our task design to improve the discriminability between these three scenarios based on behavioural data, and we performed a second experiment. Our new findings contradicted the predictions of the 'belief' and 'affect' scenarios, but were in line with those of the 'decision' scenario.

Indeed, the decision bias was observed irrespective of the reported belief in a general (in real life) correlation between effort and reward, and despite the demonstration to participants that effort and reward were independently drawn at random (in the experiment), by playing lottery animations. This finding extends previous reports that reward is used as a cue for retrospective effort judgement (*Pooresmaeili et al., 2015*; *Rollwage et al., 2018*), which was observed in experiments where reward and effort were actually correlated, and therefore interpreted as a rational Bayesian inference. Although such a Bayesian inference might be implemented in situations where it does produce valid conclusions, our results show that reward also impacts prospective effort estimates that drive decision-making even when this influence is not instrumental. This impact should therefore be seen as an irrational bias, imposed by a cognitive process that is automatically triggered despite its irrelevance in the context of the experiment. It is impossible with our design to know whether the putative cognitive process inducing a bias on effort cost estimates was triggered when receiving the information about incentive levels and running routes, when making the decision about whether to accept the offer, or when reporting subjective cost on the rating scale. By automatic, we simply mean that the bias was not (fully) controlled since it occurred despite the monetary bonus offered for accurate judgements, and despite evidence that reward magnitude and effort cost were independent from each other.

This automatic process could have involved affective reactions to incentive presentation, following on affect-as-information theory (*Clore and Storbeck, 2006*). The idea is that the brain reverses a general implication from information to affective state. For instance, anticipating an important effort goes with activating the autonomic nervous system, which is perceived as enhanced arousal level. Therefore, perceiving enhanced arousal level could signal that an important effort is to be exerted soon. We discarded this possibility by manipulating arousal level through music extracts played when participants viewed the running routes and the associated incentives. Although music did increase autonomic measures (pupil size) in proportion to arousal ratings, it did not influence effort cost estimates. Thus, the impact of incentives is unlikely to have been mediated by arousal state. However, we must acknowledge that arousal ratings were not recorded during the experiment, so it remains possible that the music was actually ineffective for modulating arousal level when participants performed the main task. A variant of this affect scenario would be that incentives modulate the affective state along a valence dimension. Participants would overestimate effort costs not because they feel more excited with higher incentives but because they feel more excited in a positive way. This mediation by valence is also unlikely because negative incentives (prospective losses) had the same impact as positive incentives (prospective gains) on effort cost estimates.

Finally, our findings are in line with the predictions of the decision scenario, which assumes that participants automatically translate reward into effort. This ability would arise from the cost–benefit trade-off machinery responsible for the motivational control of behaviour. Bayesian family model comparison across our two groups of participants provided strong evidence in favour of cost–benefit models, relative to purely descriptive, non-principled models (generalized linear regression). However, further comparison between the two variants of this hypothesis ('maximal budget' and 'optimal expense') did not suggest that one was more prevalent than the other in the sampled population of participants. This could mean that these two underlying mechanisms are equally prevalent in the population or simply that statistical power was not sufficient to detect a difference in prevalence. Besides, it is possible that both mechanisms are at play within a given individual, either at the scale of a single-effort estimate, or across several estimates, such that none of these mechanisms clearly dominates the other. In any case, we must assume that these computations are implicitly performed by the brain because participants did not report them during post-experiment debriefing interview, even when asked about a possible influence of monetary incentives on effort cost rating.

The general conclusion remains that prospective effort cost judgements are likely biased by the product of a prescriptive, decision-related cost–benefit computation. Indeed, we found higher evidence for models in which the probability of choosing the high-effort option was a function of anticipated effort cost (including distortions of the real effort cost by monetary incentives) rather than a direct function of real effort cost. This suggests that subjective distortions induced by goal value affect not only ratings of effort cost but also decisions about whether to exert more effort for a higher benefit. Since choices were made before ratings, it is unlikely that decision-making was biased because of explicit cost judgements, although one cannot exclude that participants prepared their rating beforehand. In any case, this decision bias makes ambivalent the role of goal value in expected

utility computation: on the one hand, it increases the expected benefit of the action, but on the other hand, it also increases the perceived cost of the action. The net effect of the decision bias is to make high-value goals less attractive than they should be, if participants had not anticipated that they would come at higher costs.

Although we think this work is an important step towards a mechanistic comprehension of how effort anticipation and other action-related judgements can interfere, further investigation will be needed to assess the robustness and generalizability of these findings, circumscribe the scope of this phenomenon, and gain a fine-grained understanding of its computational and neural bases. Specifically, one remaining question to be addressed is whether other types of prospective cost judgements, about mental effort in particular (but also delay or risk), are subject to the same distortions from goal value as were energetic cost judgements made about physical effort in our design. Because mental effort has also been construed as a cost traded against potential benefits in decision-making models (*Pessiglione et al., 2018*; *Shenhav et al., 2013*), we would expect the same decision bias: a cognitive activity would be anticipated as requiring more effort when it is associated to more valuable outcome. Besides, although it is unlikely that effort cost estimation occurs independently from decision-making in most real-life scenarios, it remains to be tested whether the same bias would manifest in a scenario where no choice is made and participants solely report the anticipated energetic cost. Another crucial issue regards the impact of this effect on action execution. In this study, the actual effort expended during running was controlled by the treadmill speed and slope. However, earlier studies have reported a positive effect of goal value on force production, even when it is non-instrumental or in fact detrimental to performance (*Dam, 2017*; *Oudiette et al., 2019*), which could indeed originate from a higher anticipated effort cost.

This observation leads us to interrogate the potential adaptiveness of such an influence of goal value on effort cost anticipation. We can speculate that, in environments where the invested effort and the value of the attained goal are strongly correlated, information about this goal value is on average informative about the effort that will be required. Thus, using such a heuristic might help prepare the action, in situations where value is known with precision whereas information about the cost is more uncertain. A better preparation, for an overestimated difficulty, may later result in higher persistence towards high-value goals in the face of fatigue, pain, or other forms of adversity that might be encountered during action execution. However, overestimating the cost may lead people to abandon the pursuit of high-value goals that they actually have the ability to reach. These missed opportunities may be particularly frequent in patients suffering from depression, who might be trapped in a paradoxical behaviour: ignoring attractive goals just because their high values are seemingly signalling high costs.

We acknowledge that, even if it provides a well-controlled proof of concept, the size of the effect was rather small in our data. Yet, this critically depends on the specifics of the experimental situation, such as the range of incentive values that were proposed and the fact that participants were heavily trained (and encouraged) to make accurate effort cost judgements. Further research is needed to investigate how preponderant the effect is in more naturalistic situations where people have only vague ideas about the actual costs. Another critical feature of our design is that the incentive level was clearly indicated when ratings and choices were made, while the effort cost had to be estimated and memorized while viewing the running animation. A question for future investigation is whether the decision bias might manifest in the opposite direction when information about effort cost is available for a precise subjective estimate, while information about goal value remains vague. This opposite bias would result in overestimating prospective goal value when effort cost is obviously high and would lead people to another paradox: engaging in strenuous actions just because they seem costly enough (to bring valuable reward, presumably).

## Materials and methods

The study was approved under reference C15-59 (ID RCB 2015-A01445-44) by the Ethics Committee for Biomedical Research of the Pitié-Salpêtrière Hospital, where both experiments were conducted.

### Participants

Participants were recruited via the Relais d'Information en Sciences Cognitives (RISC) website and screened for exclusion criteria: age below 18 or above 39; less than 1 hr physical exercise (and/or less

than 30 min of cardiorespiratory training) per week; regular use of drugs or medications; history of psychiatric, neurological, cardiac, or respiratory disorders.

All participants gave informed consent prior to partaking in the study.

## Experiment 1

We aimed to test a novel hypothesis, and no corresponding effect size had been previously reported in the literature. We therefore decided to use a sample size of N = 22, which typically allows to detect reasonably large behavioural effects.

Twenty-two participants (10 females, 12 males; mean age = 25.6 ± 4.7 years) were thus recruited for experiment 1. One participant could not complete the last testing block due to a technical failure, but data from the first five blocks were nevertheless included in the analyses.

Participants received an initial endowment of 18€ and earned financial bonuses during the session, leading to a final payment of 30.7€ on average (SD = 4.5€).

## Experiment 2

We used a sample size of N = 24, similar to that of experiment 1, with marginally increased statistical power.

Twenty-four participants (12 females, 12 males; mean age = 26.0 ± 5.0 years) were thus recruited for experiment 2 and were all included in the analyses.

Because participants incurred a risk of monetary loss during this experiment, they received an initial endowment of 30€, which ensured a non-negative final payment. This final payment was 41.8€ on average (SD = 4.4€).

## Apparatus

We programmed all tasks and data analyses with MATLAB (MathWorks ) and used the Psychophysics Toolbox 3.0 (*Kleiner et al., 2007*) for stimuli presentation. Participants ran on a Tech med Cardio 270 treadmill, whose speed and inclination were controlled by the computer. In experiment 2, pupil dilation was measured with a The Eye Tribe ET1000 eye-tracker (60 Hz sampling frequency) attached below the computer screen, and participants' head position was stabilized with a chin-rest placed 70 cm away from the 23″ screen. Choices and ratings were made using a standard keyboard.

## Task design

In this study, we designed a novel behavioural task aimed at measuring the influence of goal value on the anticipated energetic cost of reaching this goal.

## Experiment 1

The general idea of our experimental paradigm was as follows: each participant was shown short on-screen animations depicting a cartoon character running various routes, each associated to a varying amount of reward, and was asked to estimate a priori the energetic cost of each given route. The participant was additionally asked whether she would agree to run each given route in reality (on a treadmill) in order to earn the associated reward.

More specifically, after giving their consent, each participant completed a single 3 hr session comprising these steps:

### 1. Training
Participants went through an extensive training in order to familiarize themselves with the task.

 i. Elicitation of comfort speed

 First, they performed a 3 min physical warm-up run on the treadmill, which was pre-set at a 7 km/hr speed and a 4% slope. Then, while they kept running, they were instructed to adjust the treadmill speed to their 'comfort' level, which was described to them as the speed at which they would be able to run for 30 min without interruption nor excessive fatigue. They tuned the speed by 0.5 km/hr increments via 'plus' and 'minus' buttons and received immediate motor feedback (but no numeric feedback about the current speed was given in order to encourage them to rely on a proprioceptive rather than a strictly numeric benchmark),

until they reported having reached such a 'comfort' speed and had spent at least 2 min in the speed adjustment phase (in order to ensure a minimal warm-up duration of 5 min).

ii.  Familiarization with speed and slope levels (practice session)

Running routes which were to be displayed during the testing phase would be comprised of four segments, each a combination of 1 of 3 speed levels (110, 125, and 140% of comfort speed, expressed via the on-screen scrolling speed of dashes symbolizing the road) and 1 of 3 slope levels (0, 4, and 8%, represented by 0, 12, and 24° slopes, respectively, to make levels more distinguishable visually).

In order to allow participants to map visual, on-screen features characterizing each route segment with proprioceptive sensations (and in particular, an effort cost internal signal), we instructed them to run six 'elementary' routes successively, each corresponding to a single 1 min segment $\{Speed_i; Slope = 0\}$ then $\{Speed = S_{comfort}; Slope_i\}$ with $i \in 1; 3$. Differences in speed or slope were thus experienced with the other dimension being held constant.

iii.  Mapping global route display with energetic cost rating (calibration session)

This training phase was critical in that it allowed participants to map together, for a given route (1) its corresponding on-screen animation, (2) the physical sensations perceived while and after running this route (and associated internal effort signal), and (3) a cursor position on the energetic cost rating scale.

First, participants were instructed that their accuracy at the energetic cost estimation task would condition their final payment and were advised to picture themselves in the shoes of the running character during the animation display in order to produce accurate estimates. This was to encourage them to rely on internal simulation rather than visual cues to build their cost estimate as they would do in real-life situations. Then, they completed two trials in which they sequentially watched a running animation, ran the corresponding 2 min route on the treadmill, and were finally shown the corresponding cursor position on the energetic cost scale. Next, they completed two trials similar to the two previous ones, except that participants now tried to estimate each route's energetic cost by placing the cursor on the rating scale after having run the route and before receiving a feedback from the computer. This was followed by 12 additional trials, in which participants estimated the energetic cost directly after seeing the running animation, that is, without actually running the route, and were then given feedback about the actual energetic cost (still as a position on the rating scale), as well as numerical feedback about their estimation accuracy (e.g. 'Accuracy: 83%'), computed as the complement of the actual distance-to-target relative to the maximal possible distance-to-target: $Accuracy = 100 \left(1 - \frac{|\omega - \hat{\omega}|}{\max\{1-\omega ; \omega\}}\right)$, where $\omega$ is the actual energetic cost (on a normalized [0;1] scale) and $\hat{\omega}$ is the estimated energetic cost.

iv.  Familiarization with the main task

Before beginning the actual testing blocks, participants performed a practice block of trials, which was identical to later testing blocks except that it did not entail any real financial stake (as made clear to the participants before the beginning of the practice block).

## 2. Testing

The testing phase was composed of six blocks, each comprising 18 trials. Each trial went as follows:

i.  Display of prospects

First, the static picture of a monetary incentive was displayed on the right side of the screen for 2 s. This incentive was one of six possible amounts: {0.01, 0.10, 0.50, 1, 2, 5} € and corresponded to a bonus that would be added to the participant's final payment if she actually ran the associated route. Next, a running animation was displayed on the left side of the screen for 10 s. The animation depicted a character running along a route composed of four segments, each corresponding to a ($speed * slope$) couple, pseudo-randomly drawn from the levels described above, and lasting a pseudo-random duration (between 25 and 45 s in reality, i.e., between 2.08 and 3.75 s on-screen). An in-house algorithm ensured that the total duration of the animation was 10 s (corresponding to a real duration of 120 s), and that the actual energetic cost of the corresponding route equalled one of six predetermined energetic cost levels: $\omega \in \{.20, .32, .44, .56, .68, .80\}$ such that $EC_\omega = (1 - \omega) E_{min} + \omega E_{max}$, with $E_{min}$ the energetic cost of a 2 min run at the easiest speed and slope levels: { $110\% * S_{comfort}$;

0%}, and $E_{max}$ the energetic cost of a 2 min run at the hardest speed and slope levels: {$140\% * S_{comfort}$; 8%}. Participants were not aware that energetic costs systematically belonged to one of six possible levels. The instantaneous metabolic cost (power) of a given {Speed; Slope} couple was computed following American College of Sports Medicine's (ACSM) metabolic equation for running, which has been shown to predict actual energetic expense with good accuracy (*Hall et al., 2004*).

ii. Choice

Once the animation stopped playing, participants were presented with a choice: "Do you prefer to run this route?". Selecting the 'Yes' answer meant that they preferred running the route just displayed in order to win the monetary reward simultaneously displayed, selecting the 'No' answer meant that they preferred running a 'comfort' route (2 min at comfort speed with 0% slope) for no monetary reward. Choice was self-paced and was not executed immediately afterwards.

iii. AEC rating

Once they had validated their choice, participants were prompted to estimate the energetic cost of the running route depicted by the animation they had just watched. This self-paced rating was made by placing a cursor along a horizontal scale (implicitly bounded by $E_{min}$ and $E_{max}$). Participants were incentivized to produce accurate ratings via the following rule: if, by the end of each block, their average accuracy exceeded 80%, they would receive a 1€ bonus for this block. (Participants' average accuracy was 82.3% ± 3.8% in experiment 1.)

At the end of each block, one trial among the 18 just completed was randomly drawn by the computer; the corresponding route and the associated monetary reward were displayed, along with a reminder of the participant's choice ("You accepted/refused to run this route."). The participant was then instructed to step onto the treadmill and to run the effortful (resp. comfort) route if she had accepted (resp. refused) the offer. Upon completion of the effortful route, the associated monetary reward was added to her final payment. (Notably, all participants completed all running routes – except for one participant who chose to stop before the end of the most effortful run in the training session – so the risk dimension was negligible.)

Upon completion of the running sequence, participants were given a numerical feedback about the accuracy of their energetic cost estimates (averaged across the last 18 trials) along with a notification about their accuracy bonus ("You (do not) win a 1€ accuracy bonus for this block"). Their updated cumulative payment was also displayed ("Your current payment is 33.4€").

Before participants began the fourth block, they went through a short recalibration procedure, aimed at helping them maintaining accurate cost estimates, and consisting in two trials identical in structure to the last 12 trials of the calibration session performed during the training phase.

Since our experimental design involved 6 energetic cost levels * 6 incentive levels, the prospect matrix comprised 36 possible cost–incentive combinations, each repeated three times across the 108 trials, in a random order.

Finally, they completed a written survey and a subsequent oral debriefing with the experimenter. This allowed us to gather their general thoughts about the experiment, enquire about the heuristics or strategies they may have used to perform the task, or the influences they may have felt from the various experimental variables, and lastly, make sure that they had not guessed the real purpose of the experiment or explicitly assumed a correlation by design between the incentive and energetic cost factors of the task.

## Experiment 2

We adapted the design of experiment 2 from that of experiment 1, such that the various mechanisms we proposed to explain the effect of interest observed in experiment 1 (see interim discussion) would each make a distinct set of predictions about the behavioural outputs measured in experiment 2. We therefore introduced the following modifications to the design:

1. Each trial in the testing phase would start with a lottery animation, akin to a slot machine roll and lasting 0.8 s, in which a running route and a monetary incentive amount were shown to be randomly and independently selected on either side of the screen. The goal of this manipulation was to make the orthogonality of energetic cost and monetary incentive variables intuitive

and salient to the participants. In fact, the order in which {cost, incentive} couples would be displayed was still determined at the beginning of the session via a pseudo-random permutation performed by our computer program.

2. Additionally, we probed participants' belief in a reward => effort general implication by asking (via the computer) the following question: "Do you think that, in life, the less effort you make, the less reward you receive?", with four possible answers, ranging from 'absolutely yes' to 'not at all'.

3. Testing blocks would alternate between a gain (as in experiment 1) and a loss condition. In the gain condition, a 'yes' answer at the choice stage meant that the participant preferred running the route depicted by the animation in order to win the monetary amount displayed within the green 'GAIN' frame (subsequently referred to as the HEHG offer), rather than running the 'comfort' route (see experiment 1 description for details) for no reward (LE0G offer). In the loss condition, a 'yes' answer meant that the participant preferred running the route depicted by the animation in order not to lose the monetary amount displayed within the red 'LOSS' frame (high-effort 0-loss, or HE0L), rather than running the 'comfort' route and having this monetary amount deducted from her final payment (low-effort high-loss, or LEHL).

The gain or loss condition was displayed before each testing block began, and reminded in each trial by the coloured and labelled frame surrounding the monetary incentive picture. In order to maintain a number of three repetitions for each {energetic cost level × incentive level × gain/loss condition} combination, while keeping a session duration that would avoid excessive fatigue effects, only three incentive levels were used: {0.01, 0.5, 10} €.

During the testing phase, participants' arousal state was manipulated independently from energetic cost and incentive fluctuations via music. On each trial, once a running route and an incentive level had been selected by the lottery, and concomitantly with the running animation, a musical extract was played in the headset participants were wearing. For each {energetic cost level × incentive level × gain/loss condition} combination, three levels of arousal were tested: strongly arousing music, weakly arousing music, or no music (in blocks 3 and 4). Strongly and weakly arousing musical extracts were pseudo-randomly interleaved within each musical block.

We pre-selected 128 extracts (each 17 s long) from moderately familiar to little-known classical music pieces, and had a small panel (N = 4) rate the emotional arousal evoked by each extract. Next, we retained the 36 most and 36 least arousing excerpts based on this score, which allowed us to classify our music stimuli into a priori high vs. low arousal categories. Note that this procedure was only meant to ensure a sufficiently wide and approximately symmetrical distribution of arousal level, and that the external validity of this classification (relative to, e.g., the physiological or reported arousal measured from a large sample of the population) was not crucial to test the 'affect – arousal' hypothesis. Indeed, we additionally acquired a direct measure of the subjective arousal felt by participants in reaction to our musical extracts. Upon completion of the sixth testing block, participants listened again to each of the musical extracts they had previously heard (without any concurrent visual stimulation, in a randomized order) and were asked to report the amount of arousal they felt 'due to the musical extract', on an analogue rating scale ranging from 'none' to 'extreme'.

Pupil dilation was recorded during each arousal rating trial. In order to mitigate motion-related noise in the data, participants laid their head on a chin-rest. The luminance of all visual stimuli was roughly equalized before data acquisition and was subsequently measured with a Dr Meter LX1010B digital luxmeter, which we placed at the same position as participant's eyes in front of the computer screen, and exposed to all stimuli combinations.

## Statistical analysis

All statistical analyses were performed with MATLAB and its Statistics and Machine Learning Toolbox, as well as the Variational Bayesian Analysis (VBA) toolbox (*Daunizeau et al., 2014*).

GLMs were inverted with MATLAB's built-in *glmfit* function. All regressors were z-scored prior to GLM fitting so as to enable direct comparison of the amount of explained variance between regressors.

We subsequently tested for group-level random effects of the regressors in our models by performing one-sample two-tailed Student's *t*-tests on each vector of beta posterior estimates.

Non-linear models were inverted using a variational Bayes approach under the Laplace approximation, implemented in the VBA toolbox. The algorithm used in VBA not only inverts nonlinear models but also estimates their evidence, which represents a trade-off between accuracy (goodness of fit) and complexity (degrees of freedom).

## Experiment 1
### Choice
To assess the impact of our main experimental factors on preference for the HEHG option, we fitted a simple logistic model to binary choices made by each participant, formalized as follows:

$$P\left(ChoiceYes\right) = \frac{1}{1+e^{-\left(B_0 + k_{REC} \cdot REC + k_I \cdot I\right)}} \quad (13)$$

where $B_0$ is a bias parameter, $REC$ is the real energetic cost of the HEHG route, $I$ is the incentive (or potential reward) level of the HEHG option, and $k_{REC}$ and $k_I$ are the linear coefficients scaling the energetic cost and the incentive level, respectively.

### Anticipated energetic cost
We fitted several GLMs to the AEC ratings made by each participant.

In order to evaluate the impact of our two main experimental factors, we started with a very simple model:

$$AEC = \beta_0 + \beta_{REC} \cdot REC + \beta_I \cdot I \quad (14)$$

where $REC$ is the real energetic cost of the route, $I$ is the incentive level, and βs are the linear coefficients of our model.

Next, we tested whether the incentive level factor had an impact on AEC over and above choices made by the participant. For this, we orthogonalized our factor of interest to the potentially confounding factor:

$$AEC = \beta_0 + \beta_{REC} \cdot REC + \beta_C \cdot Choice^* + \beta_I \cdot I^* \quad (15)$$

where $Choice^*$ is the choice made by the participant (encoded as 1 for yes, 0 for no, and subsequently orthogonalized to the REC vector), and $I^*$ is the incentive level, orthogonalized to $REC$ and $Choice^*$.

## Experiment 2
### Choice
We performed a similar analysis to that in experiment 1 by inverting the following logistic model (adapted from *Equation 1* to include a loss factor):

$$P\left(ChoiceYes\right) = sigmo\left(B_0 + k_{REC} \cdot REC + k_G \cdot G + k_L \cdot L\right) \quad (16)$$

where $sigmo: x \rightarrow \frac{1}{1+e^{-x}}$, $B_0$ is a bias parameter, $REC$ is the real energetic cost of the high-effort route, $G$ is the gain level of the high-effort option (0 in loss blocks), $L$ is the loss level of the low-effort option (0 in gain blocks), and $k_{REC}$, $k_G$ and $k_L$ are the linear coefficients scaling the energetic cost, gain level, and loss level, respectively.

### Anticipated energetic cost
Again, we fitted several GLMs to the AEC ratings made by each participant.

We started with a very simple model, extended from (*Equation 2*), to include a loss factor:

$$AEC = \beta_0 + \beta_{REC} \cdot REC + \beta_G \cdot G + \beta_L \cdot L \quad (17)$$

where $REC$ is the real energetic cost of the route, $G$ is the gain level, $L$ is the loss level, and βs are the linear coefficients of our model.

Next, in order to assess the effect of the arousal evoked by the musical extracts onto AEC ratings, we fitted the following GLM:

$$AEC = \beta_0 + \beta_{REC} \cdot REC + \beta_G \cdot G + \beta_L \cdot L + \beta_A \cdot A \quad (18)$$

where $A$ is the arousal rating made by the participant about the musical extract, and $\beta_A$ is the corresponding coefficient.

## Pupil dilation

Pupil diameter time series recorded during the arousal rating task were preprocessed as follows:

1. Invalid data points (lost signal, mostly due to a blink or fixation outside the screen) were discarded, as well as all data points pertaining to a [–100; 200] ms window around these time points.
2. Outlier samples (outside three median absolute deviations from the median) were discarded.
3. A shape-preserving piecewise cubic interpolation was performed on the remaining time series.
4. An 11th-order median filtering was applied to the interpolated signal (for smoothing)
5. A $\frac{1}{128}$ Hz high-pass filter was applied to correct for low-frequency drift.
6. Time series were split into trial sequences.
7. Trial time series were baseline-corrected (by subtracting the pupil diameter averaged across the [–200;0] ms window preceding trial onset).

Next, in order to assess the effect of music-related arousal on pupil diameter, we fitted the following minimal GLM to each time point in arousal rating time series:

$$PupilDiam = \beta_0 + \beta_A \cdot A \tag{19}$$

where $A$ is the arousal rating made by the participant about the musical extract being played.

Finally, we smoothed the t-statistic time series derived from fitting this GLM by convolving them with a Gaussian kernel $(\sigma = 50\ ms)$ in order to ensure that the assumptions of the Random Field theory were met, and we applied the *RFT_GLM_contrast* function (from the VBA toolbox) to our smoothed time series. This allowed us to test the significance (in a random-effects approach) of each regressor at each time point while correcting for multiple comparison at cluster level.

## Inversion of nonlinear cost–benefit models of choice and rating

We inverted nonlinear models predicting both anticipated effort cost (AEC) rating and high-effort acceptance rate as a function of the REC, the nominal gain magnitude (G), and the nominal loss magnitude (L).

The models applied to experiment 1 data are explained in the interim discussion.

In experiment 2, the 'maximal budget' model was extended from that presented in the interim discussion to account for the presence of a loss condition. The decision variable can now be written as

$$DV = U_{HE} - U_{LE} = \left(G^\rho - C_E \cdot EC^\alpha\right) - \left(-\lambda \cdot L^\rho - C_E \cdot EC_0^\alpha\right) \tag{20}$$

where $G$ and $L$ are the prospective gain and loss magnitudes attached to the high-effort (HE) and low-effort (LE) options, respectively, $\rho$ is the nonlinearity parameter controlling the magnitude-to-utility mapping (typically, $\rho < 1$; *Glimcher and Fehr, 2013*; *Schoemaker, 1982*), $\lambda$ is a loss aversion parameter (typically, $\lambda > 1$; *Kahneman and Tversky, 1984*; *Tom et al., 2007*), $C_E$ is the scaling coefficient of the effort cost discounting variable, and $\alpha$ is the nonlinearity parameter controlling the curvature of effort discounting.

Thus:

$$DV = G^\rho + \lambda \cdot L^\rho - C_E \cdot \left(EC^\alpha - EC_0^\alpha\right) \tag{21}$$

therefore, the maximal energetic cost such that DV = 0 is

$$EC_{max} = \left(EC_0^\alpha + \frac{G^\rho + \lambda \cdot L^\rho}{C_E}\right)^{\frac{1}{\alpha}} \tag{22}$$

And the updated decision variable can be written as

$$DV' = G^\rho + \lambda \cdot L^\rho - C_E \cdot \left(AEC^\alpha - EC_0^\alpha\right) \tag{23}$$

The 'optimal expense' model was extended in a similar fashion. The expected utility from the high- and low-effort option can now be written (respectively) as

$$E\left[U_{HE}\left(EC\right)\right] = P\left(Success|EC\right) \cdot G^\rho - \left(1 - P\left(Success|EC\right)\right) \cdot \lambda \cdot L^\rho - C_E \cdot EC^\alpha$$

$$E\left[U_{LE}\left(EC\right)\right] = -\lambda \cdot L^{\rho} - C_E \cdot EC_0^{\alpha} \tag{24}$$

Such that the updated decision variable is

$$DV^{'} = P\left(Success|AEC\right) \cdot \left(G^{\rho} + \lambda \cdot L^{\rho}\right) - C_E \cdot \left(\left(AEC\right)^{\alpha} - EC_0^{\alpha}\right) \tag{25}$$

As befits a Bayesian approach, we set non-trivial prior probability distributions over model parameters. We set agnostic distributions (centred on 'null' values, e.g., $\rho$ and $\alpha$ were centred on 1) before fitting models to our first dataset (experiment 1), and used the posterior distributions derived from this first inversion stage as the prior distribution in our second inversion (experiment 2, except for the $\lambda$ parameter, which was absent from the models in experiment 1 and was attributed an agnostic prior distribution centred on 1).

## Model comparison

The log evidences, estimated for each participant in each experiment and each model via the VBA scheme, were submitted to a group-level random-effect analysis (*Rigoux et al., 2014*). This analysis was used to generate family-level protected exceedance probabilities, which quantify the likelihood that a given family of models is more frequently implemented by the population than any other family in the comparison set. This allowed us to compare the evidence for 'standard' versus 'null choice', 'cost–benefit' versus 'linear', and 'maximal budget' versus 'optimal expense' families, over and above the variants tested within each family.

# Acknowledgements

We thank Fabien Vinckier and Jean Daunizeau for their insightful comments, and Roeland Heerema for providing the music extracts.

# Additional information

### Funding

| Funder | Grant reference number | Author |
|---|---|---|
| Fondation pour la Recherche Médicale | FDT201805005808 | Emmanuelle Bioud |

The funders had no role in study design, data collection and interpretation, or the decision to submit the work for publication.

### Author contributions

Emmanuelle Bioud, Conceptualization, Data curation, Formal analysis, Funding acquisition, Investigation, Methodology, Writing – original draft, Writing – review and editing; Corentin Tasu, Data curation, Formal analysis, Investigation; Mathias Pessiglione, Conceptualization, Supervision, Funding acquisition, Validation, Investigation, Methodology, Writing – original draft, Project administration, Writing – review and editing

### Author ORCIDs

Emmanuelle Bioud http://orcid.org/0000-0003-4970-7907
Corentin Tasu http://orcid.org/0000-0002-1962-8573
Mathias Pessiglione http://orcid.org/0000-0002-6992-3677

### Ethics

The study was approved under reference C15-59 (ID RCB 2015-A01445-44) by the Ethics Committee for Biomedical Research of the Pitié-Salpêtrière Hospital, where both experiments were conducted. All participants gave informed consent prior to partaking in the study.

### Decision letter and Author response

Decision letter https://doi.org/10.7554/eLife.61712.sa1
Author response https://doi.org/10.7554/eLife.61712.sa2

## Additional files

### Supplementary files
• Transparent reporting form

### Data availability

All data generated during this study have been deposited in the Open Science Framework (OSF) under DOI: https://doi.org/10.17605/OSF.IO/57H8M.

The following dataset was generated:

| Author(s) | Year | Dataset title | Dataset URL | Database and Identifier |
|---|---|---|---|---|
| Bioud E, Tasu C, Pessiglione M | 2021 | A computational account of why more valuable goals seem to require more effortful actions | https://doi.org/10.17605/OSF.IO/57H8M | Open Science Framework, 10.17605/OSF.IO/57H8M |

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

## Appendix 1

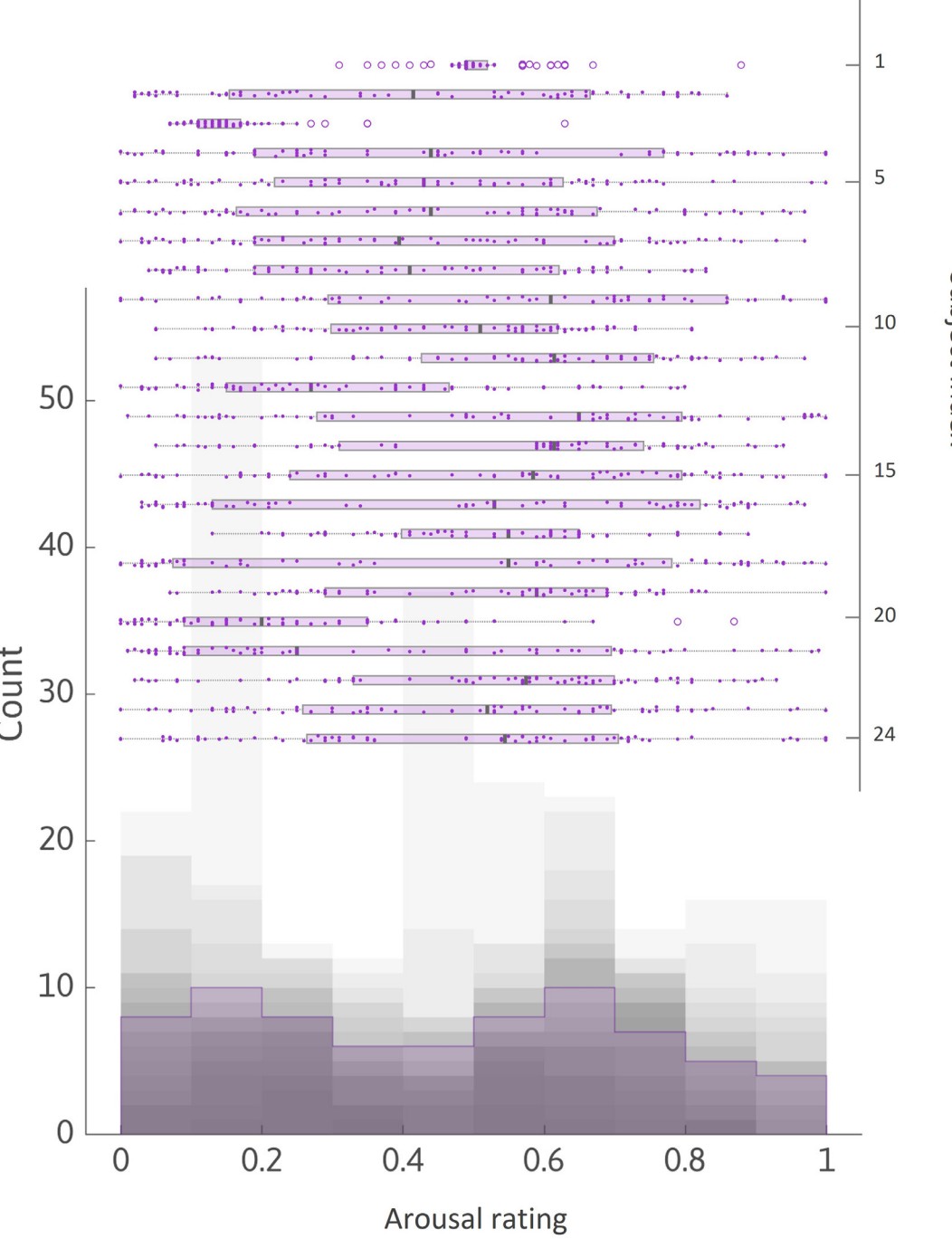

**Appendix 1—figure 1.** Distribution of arousal ratings in experiment 2. Histogram plot, left y-axis: distributions of arousal ratings made about 72 musical extracts (ratings normalized to [0; 1]) overlaid across all subjects. Grey shade indicates the number of subjects who reached a given count (or above) for each given rating bin. Purple area delineates the group-averaged distribution. Whisker plots, right y-axis: all ratings and summary statistics of each subject. Each dot represents a rating, main vertical grey bar is the median, left and right ends of boxplot are first and third quartile of the distribution, respectively.

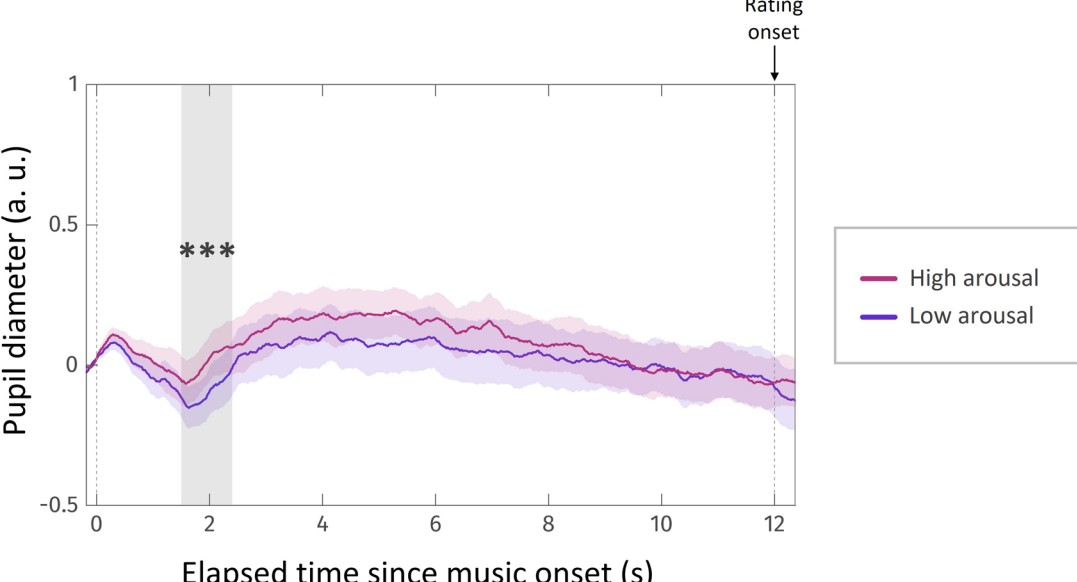

**Appendix 1—figure 2.** Pupil diameter during arousal rating task in experiment 2. Pupil diameter as a function of elapsed time since the onset of each musical extract. Time courses were median-split between trials in which a high versus low arousal rating was given. Solid lines represent time courses averaged across corresponding trials and all experiment 2 subjects. Shaded area around each curve represents the standard error to the mean (s.e.m), computed across subjects. Vertical grey area highlights the time window during which pupil diameter was larger for high-arousal extracts (time window: [1.67–2.55] s post music onset, $p = 3 \cdot 10^{-4}$, corrected for multiple comparisons). Left and right vertical dotted lines indicate music onset and rating onset, respectively.

