## [Editor Report]

This article tests a basic assumption made by many decision-making theories, specifically that costs and benefits are independent when making rational choices. As such, the findings from this study should be of interest to a wide range of researchers interested in decision-making, effort, and motivation.

---

## [Decision Letter]

**Decision letter after peer review:**

Thank you for submitting your article "A computational account of why more valuable goals seem to require more effortful actions" for consideration by *eLife*. Your article has been reviewed by 2 peer reviewers, and the evaluation has been overseen by a Reviewing Editor, David Badre, and Christian Büchel as the Senior Editor. The reviewers have opted to remain anonymous.

Essential revisions:

The reviewers agreed that this manuscript addresses an interesting and important question pertaining to a fundamental assumption of most decision-making theories. The methodical way in which alternative hypotheses were considered was noted as a particular strength. There were some points, however, that the reviewers thought should be addressed in a revision.

1. A lingering concern is that rather than separate effects of incentives on effort choice and effort ratings, it is possible that the ratings effects reflect a spillover effect of confidence from the prior effort choice. Specifically, participants' confidence in the accept/reject decisions for the effort conditions could theoretically influence the subsequent AEC rating. This prediction falls out of the following plausible conditions: (a) higher incentives are associated with more confident choices (this is based on the fact that higher incentives are associated with higher pr(accept) and there seems to be a baseline preference to accept); (b) when rating on a continuous scale, higher confidence is associated with ratings that are further from the midpoint of the scale (the authors have shown this previously, e.g., in Lebreton et al., 2015); (c) more extreme ratings (those further from the midpoint) were also associated with higher ratings of effort (this is unclear from the results presented, but seems possible given that people seemed to potentially rate effort above the midline more than below the midline). Can the authors rule out this alternative explanation? Some things to consider would be testing assumptions (a) and (c) above and examining relationships between choice accept/reject choice confidence (e.g., based on choices and RTs during that phase) and overall AEC/rating extremity (and/or associated RTs) during the AEC estimation phase.

2. The authors demonstrate that there is no interaction between incentives and ground truth on AEC. To visualize this finding and provide readers with a sense of participants' accuracy/precision in their AEC judgments, please add a figure or panel that directly visualizes the correlation between AEC and REC (on the x-axis) and how that relationship varies as a function of incentive (color coding). This figure may also be helpful in examining the extent to which there appears to be a regression to the mean/midpoint of the rating scale following low confidence accept or reject choices.

3. Rating AEC after making a decision. The maximal budget and optimal expense models assume that participants compute the AEC (estimated effort) before they make a choice about whether to select the high-effort (HE) option. However, in the experiment, participants are asked to estimate AEC *after* they make a choice. Thus, one could hypothesize that the process of making a choice may impact participants estimate of AEC but not the other way around. In other words, the AEC informing participant's choices may be a different AEC giving rise to their AEC rating. This would suggest that the choice behavior is better explained by some prior estimate of AEC (let's call this AEC_prior_decision), e.g. for the maximal budget model:

DV'=R^*ρ*−C ∙(AEC_prior_decision^*α*−EC^α_0) (Equation 5) where AEC_prior_decision could be independent of EC_max (in case of the maximal budget model) or EC* (in case of the optimal expense model), e.g.: AEC_prior_decision=B_0+B_1 * (y * REC). Note that the latter is different from the Null model in that it still explains the DV as a function of some distortion of REC (determined by B0 and B_1) but independent of EC*. The authors may want to check if the choice data (not the AEC data) is better explained by this simplified model, especially when considering the small number parameters to fit compared to the full cost-benefit models (e.g. k_0 or k_1 in the optimal expense model).

4. Ruling out belief hypothesis. The reviewers acknowledge the authors' effort in testing the belief hypothesis. However, they were not fully convinced that the measures taken are sufficient to rule out this hypothesis. First, one could cast doubt on whether the lottery manipulation actually eliminated the assumption of correlation. One could argue that participants assume instead that the lottery is biased (i.e. their belief transfers to the lottery). In addition, they were not sure if a null-effect-i.e., lack of evidence for a correlation between the behavioral effect and a single-sentence belief rating-is sufficient to rule out this hypothesis (participants needn't be aware of priors in order for those priors to affect behavior). That is, they wouldn't go so far as to say that this "contradicts one of the predictions of the 'Belief' scenario" (p. 24). This raises the question: why weren't participants explicitly told that there is no correlation between reward and effort ratings prior to the experiment? The reviewers understand that it would require another round of experimentation to incorporate such an instruction, which was not considered necessary. Instead, the authors may be able to address this issue in their discussion. Alternatively, the reviewers were wondering if the authors could formalize this hypothesis in the two decision-theoretic models where AEC does not depend on EC_max or EC* but instead on R, e.g. replace (Equation 4) with AEC = B_0 + B1 * REC + B2 * R (where B2 is constrained to be positive).

---

## [Author Response]

Essential revisions:The reviewers agreed that this manuscript addresses an interesting and important question pertaining to a fundamental assumption of most decision-making theories. The methodical way in which alternative hypotheses were considered was noted as a particular strength. There were some points, however, that the reviewers thought should be addressed in a revision.1. A lingering concern is that rather than separate effects of incentives on effort choice and effort ratings, it is possible that the ratings effects reflect a spillover effect of confidence from the prior effort choice. Specifically, participants' confidence in the accept/reject decisions for the effort conditions could theoretically influence the subsequent AEC rating. This prediction falls out of the following plausible conditions: (a) higher incentives are associated with more confident choices (this is based on the fact that higher incentives are associated with higher pr(accept) and there seems to be a baseline preference to accept); (b) when rating on a continuous scale, higher confidence is associated with ratings that are further from the midpoint of the scale (the authors have shown this previously, e.g., in Lebreton et al., 2015); (c) more extreme ratings (those further from the midpoint) were also associated with higher ratings of effort (this is unclear from the results presented, but seems possible given that people seemed to potentially rate effort above the midline more than below the midline). Can the authors rule out this alternative explanation? Some things to consider would be testing assumptions (a) and (c) above and examining relationships between choice accept/reject choice confidence (e.g., based on choices and RTs during that phase) and overall AEC/rating extremity (and/or associated RTs) during the AEC estimation phase.

This is undoubtedly a clever, sophisticated explanation.

On a conceptual level, we have reservations about point b. What has been shown, by our lab and others, is that when participants provide an extreme rating, their confidence in this rating is higher than for a medium rating. There is no evidence that higher confidence in a separate decision would have a causal influence on how extreme the rating will be.

On an empirical level, this possibility is contradicted by the observation that the positive effects of incentives on AEC is similar for low and high REC (see curves with different colors in Figure 2b and 5b). The suggested explanation would on the contrary predict that low energetic costs would be rated even lower with higher incentives, because higher confidence in the choice would make participants go closer to the lower bound of the rating scale. We have checked that when selecting the 3 lower REC levels (blue to green curves in Figure 2b), for which participants go left to the midpoint, the effect of incentives is still positive and significant, in a linear model where AEC = B0 + B1.REC + B2.Gain (+B3.Loss). This was true for prospective gain level (B2 = 1.9 ± 0.4, p < 10^-4^) in Exp. 1 and both prospective gain (B2 = 2.2 ± 0.9, p = 2.7^.^10^-2^) and prospective loss (B3 = 2.5 ± 0.9, p = 8.6^.^10^-3^) in Exp. 2.

We have also tested the suggested explanation more directly by adding a proxy for choice confidence in the previous GLM, i.e. by fitting an alternative GLM where AEC = B0 + B1.REC + B2.Confidence + B3.Gain (+ B4.Loss), with the gain and loss regressors orthogonalized to the confidence proxy. We found that both prospective gain and loss still had a significant effect (all p<3.10^-2^), while there was no effect of confidence (all p>0.29), whether we took – RT, exp(– RT) or 1/RT as a proxy.

Thus, there is no reason to believe that higher choice confidence could have made the ratings more extreme, or induced an increase in AEC rating, or mediated the effect of incentives on AEC rating.

2. The authors demonstrate that there is no interaction between incentives and ground truth on AEC. To visualize this finding and provide readers with a sense of participants' accuracy/precision in their AEC judgments, please add a figure or panel that directly visualizes the correlation between AEC and REC (on the x-axis) and how that relationship varies as a function of incentive (color coding). This figure may also be helpful in examining the extent to which there appears to be a regression to the mean/midpoint of the rating scale following low confidence accept or reject choices.

This finding was already visualized in Figure 2b for Exp. 1 and 5b for Exp. 2. It corresponds to the observation that the increase in AEC rating with incentive level is similar for the different REC levels tested (the 6 curves with different colors). The regression to the mean is visible in the fact that low REC levels are overshot (e.g., compare plain and dotted blue line), while high REC levels are undershot (e.g., compare plain and dotted red lines).

To satisfy the reviewer’s request, we have made a figure reverting the x-axis and color code (with REC on x-axis and incentive color coded). Like Fidure 2b, it shows (see Author response image 1) that the incentive effect on AEC is present for both low and high REC. The regression to the mean is visible in the fact that low REC levels (e.g., compare data points to diagonal on the left) are overshot while high REC levels (e.g., compare data points to diagonal on the right) are undershot.

**Author response image 1. sa2fig1:** 

As the two figures are strictly equivalent (they provide the exact same information), we believe there is no need to have them both in the paper. We are nonetheless happy to add the new figure in supplementary material if the reviewers find it helpful.

3. Rating AEC after making a decision. The maximal budget and optimal expense models assume that participants compute the AEC (estimated effort) before they make a choice about whether to select the high-effort (HE) option. However, in the experiment, participants are asked to estimate AEC after they make a choice. Thus, one could hypothesize that the process of making a choice may impact participants estimate of AEC but not the other way around. In other words, the AEC informing participant's choices may be a different AEC giving rise to their AEC rating. This would suggest that the choice behavior is better explained by some prior estimate of AEC (let's call this AEC_prior_decision), e.g. for the maximal budget model:DV′=R^ρ−C ∙(AEC_prior_decision^α−EC^α_0) (Equation 5) where AEC_prior_decision could be independent of EC_max (in case of the maximal budget model) or EC* (in case of the optimal expense model), e.g.: AEC_prior_decision=B_0+B_1 * (y * REC). Note that the latter is different from the Null model in that it still explains the DV as a function of some distortion of REC (determined by B0 and B_1) but independent of EC*. The authors may want to check if the choice data (not the AEC data) is better explained by this simplified model, especially when considering the small number parameters to fit compared to the full cost-benefit models (e.g. k_0 or k_1 in the optimal expense model).

Our explanation being that the process of making a choice impacts participants’ estimate of AEC, we believe that making choices before ratings is consistent with the ‘cost-benefit’ models. Yet we agree with the reviewer that, in principle, the AEC estimate used to make the decision and the AEC estimate expressed in the rating could be distorted in different ways. In the previous version, we had compared two possibilities: choice depends on objective REC (null models) versus choice depends on AEC being some linear combination of REC with incentive level (linear models) or with EC_max / EC* (cost-benefit models). We understand the above comment as suggesting an intermediate possibility: choice could depend on an affine transformation of REC. This is indeed an insightful suggestion, as the previous null models might have lost the comparison because they lacked this affine transformation and not because they missed the impact of incentives. We therefore replaced the previous null models where choice = f(REC) by new null models where choice = f(B0 + B1.REC). The results of the model comparison are unchanged (see Author response image 2): models in which choice is a function of AEC are still far more plausible than the new null models (family-wise comparison: exceedance probability > 0.99). Other results remained the same: cost-benefit models are still much more plausible than models where AEC is a linear combination of REC and incentive level, while within the cost-benefit family, the ‘maximal budget’ and ‘optimal expense’ models are still equiprobable.

Additionally, we performed a family-wise comparison between the two ‘null choice’ model families (top bars in the graph of Author response image 2), i.e. the original version with choice = f(REC) and the new version with choice = f(B0+B1.REC). The original null model was in fact more plausible, even if the comparison was technically not conclusive (exceedance probability = 0.89). We therefore prefer to keep the original version as it makes null models more challenging for our cost-benefit models. Another reason is that the affine transformation model is in fact a particular case of the linear model family (in which AEC = B0+B1.REC+B2.Incentive), with the weight on incentive being set to zero. Including this affine transformation would thus create an imbalance in the factorial model space and confuse the readability of the Bayesian model selection approach.

4. Ruling out belief hypothesis. The reviewers acknowledge the authors' effort in testing the belief hypothesis. However, they were not fully convinced that the measures taken are sufficient to rule out this hypothesis. First, one could cast doubt on whether the lottery manipulation actually eliminated the assumption of correlation. One could argue that participants assume instead that the lottery is biased (i.e. their belief transfers to the lottery). In addition, they were not sure if a null-effect-i.e., lack of evidence for a correlation between the behavioral effect and a single-sentence belief rating-is sufficient to rule out this hypothesis (participants needn't be aware of priors in order for those priors to affect behavior). That is, they wouldn't go so far as to say that this "contradicts one of the predictions of the 'Belief' scenario" (p. 24). This raises the question: why weren't participants explicitly told that there is no correlation between reward and effort ratings prior to the experiment? The reviewers understand that it would require another round of experimentation to incorporate such an instruction, which was not considered necessary. Instead, the authors may be able to address this issue in their discussion. Alternatively, the reviewers were wondering if the authors could formalize this hypothesis in the two decision-theoretic models where AEC does not depend on EC_max or EC* but instead on R, e.g. replace (Equation 4) with AEC = B_0 + B1 * REC + B2 * R (where B2 is constrained to be positive).

We fully agree with this comment.

Fortunately, the experimentation suggested by the reviewers, incorporating the explicit instruction that reward and effort levels were independent, was precisely the experiment conducted and reported as experiment 2 in the manuscript. We apologize for not having made that clear, but participants were indeed told that there was no relationship between reward and effort levels, which were independently drawn before every trial. The lottery animation was implemented to ensure that participants remembered this instruction all along. This is now explicitly stated both in the methods sections and in the transition to Exp. 2, as follows (page 19):

“Under the “belief” scenario, we reasoned that if the orthogonality between monetary incentive and effort cost was made clear to participants, we should lose the significant impact of prospective reward on anticipated energetic cost. We therefore explained to participants that effort and reward levels associated to running routes were independently and randomly selected. We also made this orthogonality intuitive and salient by introducing a lottery animation at the beginning of each trial. This virtual lottery depicted the roll of a slot machine, where a running route was (randomly) selected in the left slot and a monetary amount was selected (independently) in the right slot.”

Note that the question asked during debriefing was not about the effort-reward correlation in the task (which would be trivially asking participants to repeat the instruction) but about effort-reward correlation in life (which assessed their general prior about this correlation). Yet we agree that the absence of correlation with the decision bias is not strong evidence, so we have toned down the conclusion drawn for this particular argument. However, when put together with the fact that the same decision bias was observed, whether the absence of correlation was highlighted or not, makes the ‘belief scenario’ unlikely, in our opinion.

The suggestion made to formalize this hypothesis, with AEC depending on R instead of EC_max or EC* was actually implemented in our analysis. It was actually our starting point (see Equation 2’ for Exp. 1 and Equation 5’ for Exp. 2), i.e. the simplest account of the reward bias we could imagine. This simple formalization was included in the model comparison as the ‘linear family’.

Results showed that this ‘linear family’ was far less plausible than the ‘costbenefit family’ (see author response image 2), where AEC depends on EC_max or EC*, and not directly on R.